# Synthetic mapping of XCO₂ retrieval performance from shortwave infrared measurements: impact of spectral resolution, signal-to-noise ratio and spectral band selection

Matthieu Dogniaux[1,*], Cyril Crevoisier[1]

[1]Laboratoire de Météorologie Dynamique/IPSL, CNRS, École polytechnique, Institut Polytechnique de Paris, Sorbonne Université, École Normale Supérieure, PSL Research University, 91120 Palaiseau, France
*now at: SRON Netherlands Institute for Space Research, Leiden, The Netherlands

*Correspondence to*: Matthieu Dogniaux (M.Dogniaux@sron.nl)

**Abstract.** Satellites have been providing spaceborne observations of the total column of $CO_2$ (noted $X_{CO_2}$) for over two decades now and, with the need for independent verification of Paris Agreement objectives, many new satellite concepts are currently planned or being studied to complement or extend the already existing instruments. Depending on whether they are targeting natural and/or anthropogenic fluxes of $CO_2$, the design of these future concepts vary greatly. The characteristics of their shortwave infrared (SWIR) observations notably explore several orders of magnitude in spectral resolution (from $\lambda/\Delta\lambda{\sim}400$ for Carbon Mapper to $\lambda/\Delta\lambda{\sim}25000$ for MicroCarb) and include different selections of spectral bands (from one to four bands, among which the $CO_2$-sensitive 1.6 µm and/or 2.05 µm bands). Besides, the very nature of the spaceborne measurements is also explored: for instance, the NanoCarb imaging concept proposes to measure $CO_2$-sensitive truncated interferograms, instead of infrared spectra as other concepts, in order to significantly reduce the instrument size. This study synthetically explores the impact of three different design parameters on $X_{CO_2}$ retrieval performance, as obtained through Optimal Estimation: (1) the spectral resolution; (2) the signal-to-noise ratio (SNR) and (3) the spectral band selection. Similar performance assessments are completed for the exactly-defined OCO-2, MicroCarb, Copernicus $CO_2$ Monitoring (CO₂M) and NanoCarb concepts. We show that improving SNR is more efficient than improving spectral resolution to increase $X_{CO_2}$ precision when perturbing these parameters across two orders of magnitude, and that low-SNR and/or low spectral resolution yield $X_{CO_2}$ with vertical sensitivities giving more weight to atmospheric layers close to the surface. The exploration of various spectral band combinations illustrates, especially for lower spectral resolutions, how including an $O_2$-sensitive band helps to increase optical path length information, and how the 2.05 µm $CO_2$-sensitive band contains more geophysical information than the 1.6 µm band. With very different characteristics, MicroCarb shows a $CO_2$ information content only slightly higher than CO₂M, which translates into lower $X_{CO_2}$ random errors, by a factor ranging from 1.1 to 1.9 depending on the observational situation. The NanoCarb performance for a single pixel of its imager compares to concepts that measure spectra at low-SNR and low-spectral resolution but, as this novel concept would observe a given target several times during a single overpass, its performance improves when combining all the observations. Overall, the broad range of results obtained through this synthetic

$X_{CO_2}$ performance mapping hints at the future intercomparison challenges that the wide variety of upcoming $CO_2$-observing concepts will pose.

## 1 Introduction

Anthropogenic emissions of carbon dioxide ($CO_2$) are the main driver of climate of change (IPCC, 2021). The current understanding of the global carbon cycle is based on comparisons of results from bottom-up methods, that explicitly model $CO_2$-emitting and absorbing mechanisms, with those from top-down approaches, that rely on a set of $CO_2$ atmospheric concentration observations to find the $CO_2$ fluxes that best fit those observations (Friedlingstein et al., 2022). This last approach, called inverse atmospheric transport (Ciais et al., 2010), can ingest in-situ observations and/or space-borne remote estimations of $CO_2$ atmospheric concentration. The latter are produced through inverse radiative transfer that enables to find the atmospheric states (among which the $CO_2$ concentration) that best fit infrared satellite measurements made from space.

Shortwave infrared (SWIR) satellite measurements, which are sensitive – among others – to $CO_2$ concentration close to the surface, where fluxes take place, have now been exploited for two decades to retrieve the column-averaged dry-air mole fraction of $CO_2$ (also called 'total column' and noted $X_{CO_2}$). The pioneering ESA Scanning Imaging Absorption Spectrometer for Atmospheric Chartography (SCIAMACHY) instrument (Bovensmann et al., 1999) was the first to provide a global $X_{CO_2}$ dataset. Its mission ended in 2012, and it was followed by the – still flying – JAXA/NIES Greenhouse gases Observing SATellites (GOSAT, Inoue et al., 2016; Noël et al., 2021; Taylor et al., 2022), NASA Orbiting Carbon Observatory-2 and -3 (OCO-2 and -3, Taylor et al., 2023) and the Chinese TanSat (Yang et al., 2020). The global $X_{CO_2}$ datasets produced by these missions have found applications for the study of natural carbon fluxes at global scale (e.g. Chevallier et al., 2019; Peiro et al., 2022) and also, even if it was not their primary ambition, for the monitoring of point-source anthropogenic emissions (Nassar et al., 2021; Reuter et al., 2019; Zheng et al., 2020).

These different missions will be followed by various concepts that are already planned or still being studied. First, the planned CNES MicroCarb mission (Bertaux et al., 2020; Pascal et al., 2017) is quite similar to OCO-2 regarding its observation strategy (spatial and spectral resolution, see Table 1, it includes an extra $O_2$-sensitive band) and mainly aims to provide information on natural $CO_2$ fluxes. The 2015 Paris Agreement and the five-year global stocktake system it set up have put in motion a global ambition for spaceborne monitoring of anthropogenic greenhouse gas emissions (mainly for $CO_2$ and methane, but the latter is not the focus of this work). Indeed, urban areas, that account for 0.5% of the ice-free continental surface (Liu et al., 2020; Lwasa et al., 2022), are responsible for 70% fossil fuel-related emissions (Duren and Miller, 2012). For favourable meteorological conditions, $CO_2$ plumes arise from either hotspots, such as megacities, or point sources, such as coal-fired power plants (Kuhlmann et al., 2019). Those may then be observed with SWIR spaceborne imagers, depending on their precision and spatial resolution, and the emission rate associated with the imaged plume can then be inferred either with plume

analysis/mass-balance approaches (Bovensmann et al., 2010; Varon et al., 2018) or within more usual atmospheric inversion schemes (Broquet et al., 2018; Pillai et al., 2016). Because an infrared detector has a limited number of pixels, future - planned or studied - $CO_2$ imaging concepts explore various trade-offs between spatial and spectral resolution, spectral band selection, and even compromise with the very nature of the measurements made by the instrument, when other constraints such as size and thus costs are taken into consideration. These concepts – not exclusively – include the European Copernicus $CO_2$ Monitoring ($CO_2M$ Meijer, 2020) mission, the Japanese Global Observing SATellite for Greenhouse gases and Water cycle (GOSAT-GW, Matsunaga and Tanimoto, 2022), the American non-profit Carbon Mapper initiative (https://carbonmapper.org/) based on the Next-Generation of NASA Airborne Visible/Infrared Imaging Spectrometer (Cusworth et al., 2021; Hamlin et al., 2011), the German CO2image concept (Strandgren et al., 2020; Wilzewski et al., 2020) or the European Space CARBon Observatory (SCARBO) H2020 concept, that does not measure spectra but only truncated interferograms (Brooker, 2018; Dogniaux et al., 2022; Gousset et al., 2019). Table 1 gathers the characteristics of upcoming or studied SWIR $CO_2$ observing satellite concepts, provided either in scientific articles (in this case citations are provided), in conference presentations (just the conference name and dates are given), or websites (just the hyperlink is given), as some of these concepts are quite recent.

**Table 1. Measurement characteristics for some of the upcoming or studied SWIR $CO_2$ observing satellite concepts**

| Concept | Spatial resolution/swath | Spectral bands | Resolving power ($\lambda/\Delta\lambda$) | Reference |
|---|---|---|---|---|
| **OCO-2** | 1.3x2.3 km$^2$/10 km | $O_2$: 0.76 μm | ~18000 | (Crisp et al., 2017) |
| | | $CO_2$: 1.6 μm | ~19800 | |
| | | $CO_2$: 2.05 μm | ~19800 | |
| **MicroCarb** | 4.5x8.9 km$^2$/13.5 km | $O_2$: 0.76 μm | ~25400 | (Bertaux et al., 2020) |
| | | $CO_2$: 1.6 μm | ~25750 | |
| | | $CO_2$: 2.05 μm | ~25800 | |
| | | $O_2$: 1.27 μm | ~25800 | |
| **$CO_2M$** | 2x2 km$^2$/>250 km | $O_2$: 0.76 μm | ~6300 | (Meijer, 2020) |
| | | $CO_2$, $CH_4$: 1.6 μm | ~5400 | |
| | | $CO_2$: 2.05 μm | ~5800 | |
| **GOSAT-GW** | 3x3 km$^2$/90 km and 10x10 km$^2$/920 km | $O_2$: 0.76 μm | >14000 | IWGGMS-17, 14$^{th}$ – 17$^{th}$ of June, 2021 |
| | | $CO_2$, $CH_4$: 1.6 μm | >8000 | |
| **CO2image** | 50x50 m$^2$/50 km | $CO_2$: 2.05 μm | ~1600 | (Strandgren et al., 2020) |
| **Carbon Mapper** | 30x30 m$^2$/18 km | 0.4 – 2.5 μm | ~400 around 2.05 μm | https://carbonmapper.org/our-mission/technology/ |

| SCARBO | 2.3x2.3 km$^2$/195.5 km | Truncated interferograms sensitive to:<br>$O_2$: 0.76 µm<br>$CO_2$, $CH_4$: 1.6 µm<br>$CO_2$: 2.05 µm | (Brooker, 2018; Dogniaux et al., 2022; Gousset et al., 2019) |

The characteristics of an observing concept (nature of measurement, spectral resolution, spectral band selection, signal-to-noise ratio) translate into an $X_{CO_2}$ retrieval performance that comprises (1) random error, (2) systematic error, and (3) vertical sensitivity. First, $X_{CO_2}$ random error (or precision) impacts the a posteriori uncertainties of fluxes estimated in usual inverse atmospheric schemes (Rayner and O'Brien, 2001), and the detectability of $CO_2$ plumes for imaging concepts (Kuhlmann et al., 2019). In addition to random errors, systematic errors can hamper $X_{CO_2}$ retrievals. Those can for example be due to forward

radiative transfer modelling errors, like aerosol misknowledge (Houweling et al., 2005; Reuter et al., 2010), or a priori misknowledge of atmospheric state parameters (Connor et al., 2008). Spatially correlated systematic errors are especially detrimental in inverse atmospheric schemes (Broquet et al., 2018; Chevallier et al., 2007), whereas scene-wide systematic errors that do not correlate with the plume shape can cancel out when applying plume analysis techniques. Finally, the retrieved $CO_2$ total columns must be characterized by their vertical sensitivity, which illustrates to which atmospheric levels retrievals

are sensitive (Boesch et al., 2011; Buchwitz et al., 2005).

The impact of SWIR measurement characteristics on $X_{CO_2}$ retrieval performance have been partially examined in previous studies that relied on real measurements. For instance, Galli et al. (2014) assessed the performance of $X_{CO_2}$ retrievals from GOSAT measurements which spectral resolution was degraded up to 6 times ($\lambda/\Delta\lambda \sim 3000 - 20000$), or Wu et al. (2020)

performed a similar exercise with OCO-2 measurements degraded at CO2M spectral resolution. Spectral band selection has also been studied: Wu et al. (2019) performed retrievals only using the 2.05 µm band of OCO-2 measurements and Wilzewski et al., (2020) considered single-band observations, from spectrally degraded 1.6 µm / 2.05 µm GOSAT band measurements ($\lambda/\Delta\lambda \sim 700 - 8100/6150$, respectively).

In this work, we perform a systematic survey that synthetically explores the impact of spectral resolution, signal-to-noise ratio (SNR) and spectral band selection, three design parameters for SWIR $CO_2$ observing satellite concepts, on $X_{CO_2}$ retrieval performance (SNR-related precision, degrees of freedom, vertical sensitivity and smoothing error, excluding accuracy). These choices are motivated by the characteristics gathered in Table 1. Indeed, two orders of magnitude in resolving power ($\lambda/\Delta\lambda$) separate Carbon Mapper (AVIRIS-NG) from MicroCarb. Exploring a wide range of SNR values on the top of different

resolving powers will help to encompass all possible performance results from a wide variety of concepts that measure SWIR spectra. Finally, because CO2image is planned to only measure the 2.05 µm band, and GOSAT-GW the 0.76 µm and 1.6 µm bands, we will also study the impact of choosing different combinations of spectral bands. Synthetic calculations performed

for a fictious concept with varying design parameters will help to map a large space of possible $X_{CO_2}$ retrieval performances, to which those of the peculiar SCARBO concept will be compared, along with those of the current OCO-2 and upcoming MicroCarb and CO2M missions.

This article is structured as follows: Section 2 describes the observing concepts considered in this work, and Section 3 details the materials and methods. Section 4 describes the results obtained for a fictious concept with varying design parameters and discusses them. It first focuses on the impact of spectral resolution and SNR, then on the impact of spectral resolution and spectral band selection, for which it also explores geophysical information entanglements. Finally, Section 5 discusses the performance results obtained for the exactly defined OCO-2, MicroCarb and CO2M concepts, along with those of the peculiar NanoCarb concept, and how they compare to those of the fictious concept with varying design parameters. Section 6 highlights the conclusions of this work.

## 2 SWIR CO₂ observing satellite concepts

In this section, we provide the measurement characteristics that are used to model the different upcoming – real or fictitious – SWIR CO₂ observing satellite concepts. In order to reduce the number of dimensions to explore, we consider, for the purpose of this study, that the spectra-measuring concepts have an identical resolving power $\lambda/\Delta\lambda$ over all their spectral bands, as well as a constant spectral sampling ratio of 3 (which is the case for both MicroCarb and CO₂M, and is a design hypothesis for CO2image). All Instrument Spectral Response Functions (ISRFs) are treated as Gaussian functions with a Full Width at Half-Maximum (FWHM, $\Delta\lambda$) calculated from the resolving power $\lambda/\Delta\lambda$, with $\lambda$ being the average spectral band wavelength.

### 2.1 OCO-2, MicroCarb and CO₂M

We consider three explicitly described upcoming concepts that measure and will measure SWIR spectra: OCO-2, MicroCarb and the Copernicus CO₂ Monitoring (CO₂M) concept. The left panel of Fig. 1 illustrates MicroCarb and CO₂M observations.

The Orbiting Carbon Observatory-2 (OCO-2) has been providing $X_{CO_2}$ observations from SWIR measurements for close to a decade (Taylor et al., 2023). We include this instrument in order to assess how the synthetic results obtained here relate to results obtained from real data. We model OCO-2 observations relying on instrument functions and noise models provided in OCO-2 L1b Science and Standard L2 products of Atmospheric Carbon Observation from Space algorithm version 8 (ACOS, O'Dell et al., 2018). These files are not from the latest v10 version of OCO-2 data, but the v8 to v10 major reprocessing did not include significant changes on instrument parameters (Taylor et al., 2023), so we assess that our input data are acceptable for this synthetic study.

MicroCarb (Bertaux et al., 2020; Pascal et al., 2017) is the upcoming CNES $CO_2$ observing mission that will acquire SWIR spectra at high spectral resolution, thus following the steps of the currently flying OCO-2. Besides the increase in spectral resolution, its main novelty is the addition of the $O_2$ 1.27 µm band that will provide additional optical path length information at wavelengths closer to those that contain $CO_2$ sensitivity, which may help to reduce aerosol-related errors. MicroCarb aims at retrieving $X_{CO_2}$ with a precision below 1 ppm, and with the lowest possible systematic errors. In this work, we use the measurement characteristics presented in Table 2 to model the MicroCarb concept. In Sect. 5, where MicroCarb results are presented, the impact on performance of using both or just one of the $O_2$-sensitive will be discussed.

**Table 2. MicroCarb measurement characteristics used in this work.**

| Spectral band | 1 ($O_2$ A-band) | 2 ($CO_2$ weak band) | 3 ($CO_2$ strong band) | 4 ($O_2$ 1.27 µm band) |
|---|---|---|---|---|
| Wavelenghts (µm) | 0.758 – 0.769 | 1.597 – 1.619 | 2.023 – 2.051 | 1.265 – 1.282 |
| Resolving power ($\lambda/\Delta\lambda$) | 25000 | 25000 | 25000 | 25000 |
| Spectral sampling ratio | 3 | 3 | 3 | 3 |
| Reference radiance $L_{ref}$ (W/m$^2$cm$^{-1}$sr) | 4.38 x 10$^{-3}$ | 2.69 x 10$^{-3}$ | 9.95 x 10$^{-4}$ | 2.97 x 10$^{-3}$ |
| Reference SNR $SNR_{ref}$ | 480 | 579 | 249 | 503 |

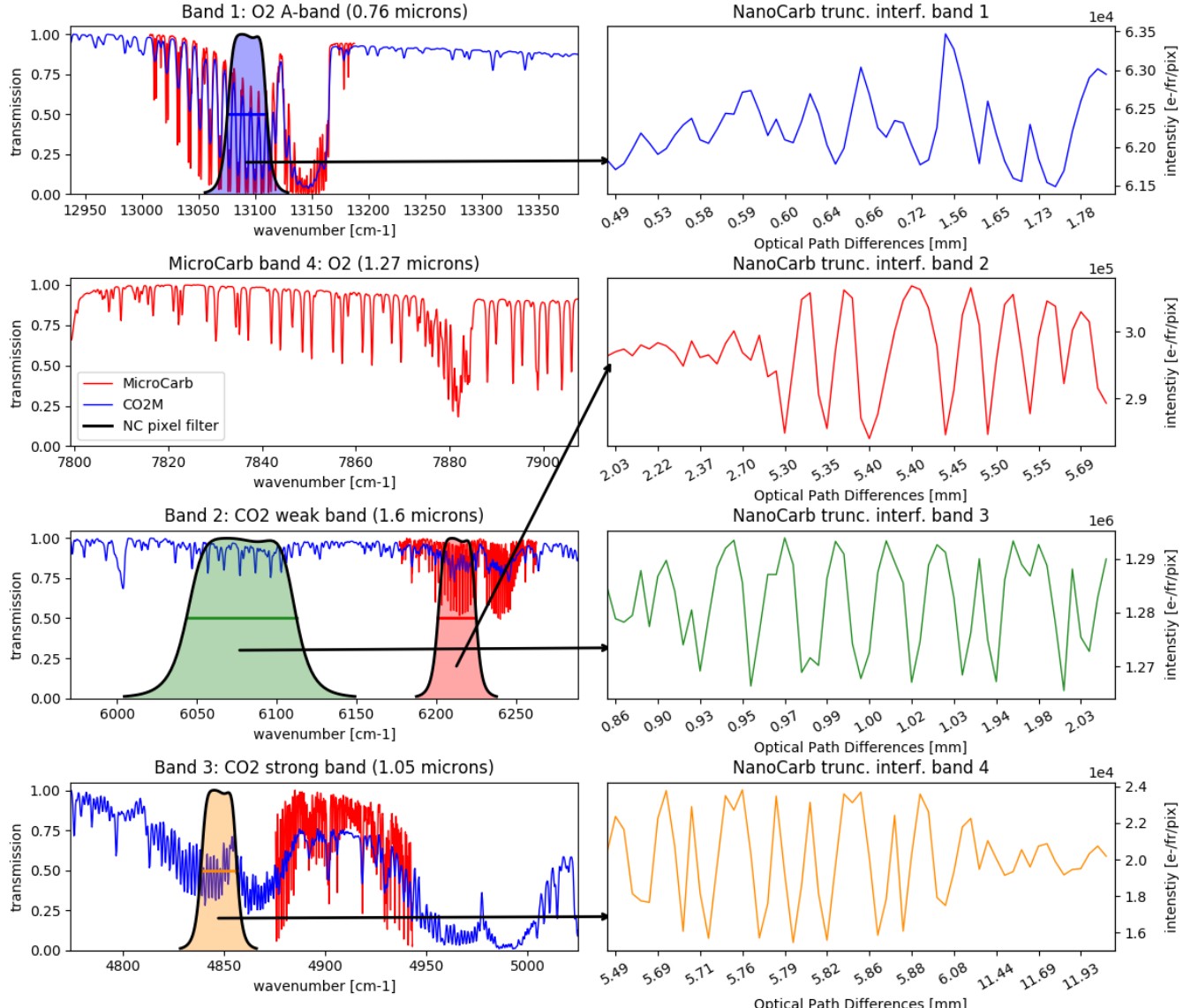

**Figure 1. Example of CO₂M (blue) and MicroCarb (red) transmissions (left) and NanoCarb truncated interferogram (right) for a vegetation-like albedo with a Solar Zenith Angle of 50°. Arrows link NanoCarb bands to their respective narrow-band filters (the horizontal coloured lines denoting their FWHMs), shown over CO₂M and MicroCarb transmissions.**

The Copernicus $CO_2$ Monitoring (CO₂M) mission (Meijer, 2020) is the upcoming space component of the operational European anthropogenic $CO_2$ emissions Monitoring and Verification Support (CO₂MVS) capacity (Janssens-Maenhout et al.,

2020). Its design compromises between spatial and spectral resolutions, swath and spectral band width, aiming to provide an imaging of $X_{CO_2}$ with a random error lower than 0.7 ppm and systematic errors as low as possible (Meijer, 2020). In this work,

we use the spectrometer measurement characteristics presented in Table 3 to model the $CO_2M$ concept. Besides the spectrometer, the $CO_2M$ mission will also include a Multi-Angle Polarimeter, which is an instrument dedicated to the observation of aerosols. Its results are expected to help better constrain their interfering effect on $X_{CO_2}$ retrievals, and improve

their precision and accuracy (Rusli et al., 2021). Here, we only study the $CO_2M$ spectrometer alone, thus the results that we obtain do not reflect the comprehensive theoretical $CO_2M$ mission performance.

**Table 3. $CO_2M$ spectrometer measurement characteristics used in this work.**

| Spectral band | 1 ($O_2$ A-band) | 2 ($CO_2$ weak band) | 3 ($CO_2$ strong band) |
|---|---|---|---|
| Wavelenghts (µm) | 0.747 – 0.773 | 1.590 – 1.675 | 1.990 – 2.095 |
| Resolving power ($\lambda / \Delta\lambda$) | 5870 | 5870 | 5870 |
| Spectral sampling ratio | 3 | 3 | 3 |
| Reference radiance $L_{ref}$ (W/m²cm⁻¹sr) | $9.66 \times 10^{-4}$ | $6.81 \times 10^{-4}$ | $7.30 \times 10^{-4}$ |
| Reference SNR $SNR_{ref}$ | 330 | 400 | 400 |

### 2.2 Fictitiously Varying $CO_2M$ concept (CVAR)

In order to grasp the full extent of upcoming or studied SWIR $CO_2$ observing satellite concepts, we also consider a fictitious concept that has the same characteristics as $CO_2M$, apart from its resolving power $\lambda / \Delta\lambda$, SNR and spectral band selection. This varying concept will be hereafter referred to as "CVAR".

First, we consider resolving power values ranging from 200 to 30000 (the list of exact resolving power values that are considered is given in the Supplementary Table S1). Figure 2 illustrates the impact of spectral resolution on the $CO_2$ absorption band around 1.6 µm. For the lowest resolving power $\lambda / \Delta\lambda = 200$, the "two-lobed" P-R branch structure of this $CO_2$ absorption band (Liou, 2002) is not visible. It fully appears from $\lambda / \Delta\lambda = 1000$ upwards. Individual absorption lines become visible, but are not fully resolved for $\lambda / \Delta\lambda$ comprised between about 1000 and 10000. Only for $\lambda / \Delta\lambda > 10000$ does the whole P-R band

structure and individual absorption lines fully appear. Given that the fixed $CO_2M$ spectral band intervals are quite large, compared to those of MicroCarb, the choice of using $CO_2M$ band intervals for exploring the impact of the resolving power on $X_{CO_2}$ retrieval performance is a reasonable compromise between high resolution instruments that measure narrow spectral bands (e.g MicroCarb or OCO-2), and low resolution instruments that measure continuous spectra (e.g. Carbon Mapper from

0.4 to 2.5 µm). Thus, this compromise yields fictitiously large spectral bands for CVAR cases with high resolving power
values, and corresponds to a window selection approach for observations with low spectral resolution, similar to what is
actually done to process AVIRIS-NG measurements (Cusworth et al., 2021).

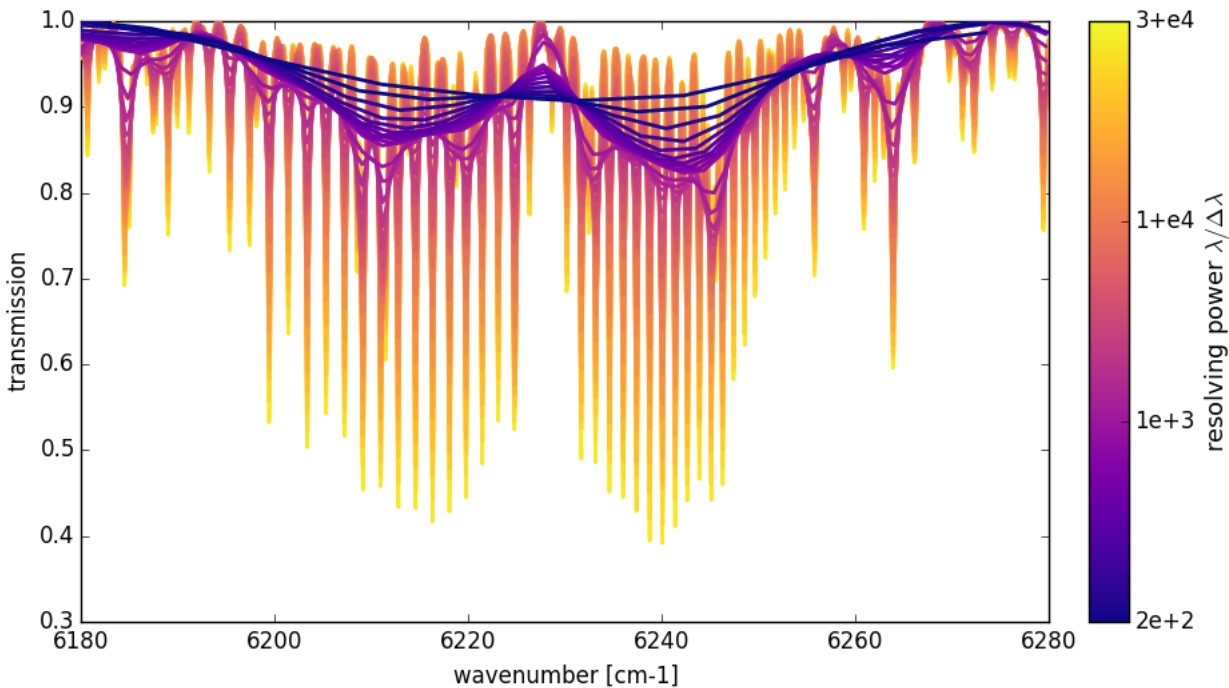

**Figure 2. $CO_2$-sensitive 1.6 µm band observed with a resolving power $\lambda/\Delta\lambda$ ranging from 200 to 30000.**

In addition to spectral resolution, we also consider the impact of SNR in this study. This will help to explore the performance
of a wider range of SWIR $CO_2$ observing satellite concepts. We will cover two orders of magnitude in noise level by applying
a spectral-band-wise factor ranging from 0.1 to 10 on CO2M SNR values given in Table 3. The impacts of both spectral
resolution and SNR on $X_{CO_2}$ retrieval performance results are presented and discussed in Sect. 4.1.

Finally, in addition to spectral resolution but separately from SNR, we consider the impact of spectral band selection. This will
help to encompass upcoming or studied single- or dual-band observing concepts such as CO2image or GOSAT-GW,
respectively. All CO2M spectral band combinations containing at least one $CO_2$-sensitive band will be explored: B2, B12, B3,
B13, B23 and B123 (with B denoting "band" followed by the CO2M spectral band numbers considered in the combination).
The impacts of both spectral resolution and spectral band selection on $X_{CO_2}$ retrieval performance results and geophysical
information entanglement are presented and discussed in Sect. 4.2 and 4.3.

## 2.3 The SCARBO concept and NanoCarb

The Space CARBon Observatory (SCARBO) concept (Brooker, 2018) is quite different from all the other concepts mentioned in this article. Indeed, it relies on a miniaturized static Fabry-Perot interferometer, named NanoCarb, that measures truncated interferograms at Optical Path Differences (OPDs) which are optimally sensitive to $CO_2$, and to some other interfering geophysical variables (Gousset et al., 2019). Because of their very nature, these truncated interferograms are sensitive to the periodic signature of $CO_2$ in the infrared spectrum. As in Dogniaux et al. (2022), we use here the latest design of the NanoCarb instrument, currently considered with a ~200 km swath and a 2.3 x 2.3 km² spatial resolution. The optimal OPD selection accounts for $CO_2$ information entanglements with $H_2O$ and aerosols, neglects atmospheric temperature and assumes that albedo is band-wise constant (Gousset et al., 2019). It measures truncated interferograms that are sensitive to four spectral bands, as shown in Fig. 1 right panel, which are associated to narrow-band filters shown in Fig. 1 left panel (follow the arrows). Their FWHMs are 35 cm$^{-1}$, 24 cm$^{-1}$, 69 cm$^{-1}$ and 18 cm$^{-1}$ for band 1 to 4, respectively. NanoCarb has a two-dimensional field-of-view (FOV, 170 across-track x 102 along-track pixels, in the current design) that observes a fixed location on the ground with different viewing angles, as it flies over it. The up to 102 $X_{CO_2}$ retrievals that can be done for a same location on the ground are then combined to yield only one unique retrieval result, with reduced random error (assuming independent observations). Dogniaux et al. (2022) details the NanoCarb concept performance results and its current shortcomings. One of the main results is that NanoCarb performance decreases close to the FOV edges, so we will only focus here on the central FOV pixel, and central along-track row of pixels. Besides, it also explains that $CO_2$ and interfering geophysical variable information contents are entangled in NanoCarb truncated interferograms. This specific shortcoming will be further detailed in this article.

## 3 Materials and methods

### 3.1 Atmospheric and observational situations

As this study focuses on the impact of instrument design parameters on $X_{CO_2}$ retrieval performance, we purposefully limit the number of atmospheric conditions that we include. We consider 12 atmospheric and observational situations that explore three surface albedo models (soil, vegetation and desert, denoted SOL, VEG and DES, respectively – their average values over the SWIR spectral bands are given in the Supplementary Table S2), generated from the ASTER spectral library (Baldridge et al., 2009), and 4 Solar Zenith Angles (hereafter SZA, 0°, 25°, 50° and 70°). A given situation will be referred to with its albedo model short name followed by the SZA, e.g. VEG-50°, for the situation with an albedo representative of vegetation, lit with an SZA equal to 50°.

For these 12 situations, the measurements are made at nadir (viewing zenith angle equal to 0°). We use a typically European atmospheric situation (vertical temperature and water vapour profiles), taken as the average of the mid-latitude temperate atmospheric profiles included in the Thermodynamic Initial Guess Retrieval (TIGR) climatology library (Chedin et al., 1985;

Chevallier et al., 1998). For this synthetic performance study, we consider a constant vertical $CO_2$ concentration profile of 394.95 ppm. The surface pressure is constant and set at 1013 hPa. To mimic possible pollution over the European continent we include fine-mode aerosols, representative of soot, between 0 and 2 km of altitude, and coarse mode aerosols, representative of minerals, between 2 and 4 km of altitude (this choice is supported by transported desert dust layers over Europe described by Papayannis et al. (2008).

## 3.2 Performance evaluation with Optimal Estimation

### 3.2.1 General aspects

Optimal Estimation (Rodgers, 2000) offers an ideal framework for the evaluation of $X_{CO_2}$ retrieval performance. It has been extensively described in other publications (e.g. Connor et al., 2008), so only its essential aspects are reminded in this article. Given a state vector $x$ that contains parameters that describe the atmospheric and surface state, and a measurement vector $y$ that contains the infrared observation made from space by a studied concept, OE enables to provide the geophysical state that best fits the measurement made from space, thus giving a satisfying solution to the following equation:

$$y = F(x) + \varepsilon \tag{1}$$

with $F$, the forward radiative transfer model that allows to simulate spaceborne infrared observations from geophysical state parameters, and $\varepsilon$, the spaceborne measurement uncertainty. Because this inverse problem is ill-posed, OE brings in a priori information that helps to better constrain the estimation. This a priori information can be seen as the knowledge of the geophysical state one would have before using the information contained in the spaceborne measurement (e.g. taken from climatologies). It is given in the form of an a priori state vector $x_a$, characterized by its uncertainty given in the a priori state covariance matrix $S_a$.

The retrieved geophysical state that best fits the measurement made from space and the a priori information is called the maximum-likelihood a posteriori state and is noted $\hat{x}$. Its a posteriori covariance matrix, which describes the uncertainty of the retrieved state, is noted $\hat{S}$ and computed using the following equation:

$$\hat{S} = [S_a^{-1} + K^T S_e^{-1} K]^{-1} \tag{2}$$

with $S_e$, the a priori covariance matrix of the measurement vector describing measurement/forward modelling uncertainties, and $K$, the Jacobian matrix containing the partial derivatives of the measurement with respect to the state vector parameters. Its elements are illustrated in the Supplements (see Fig. S1 and S2), for a usual SWIR spectrum and corresponding NanoCarb truncated interferogram.

Another useful OE result is the averaging kernel matrix, denoted $A$, which describes how the retrieved state $\hat{x}$ relate to the true – but unknown – geophysical state:

$$A = \frac{\partial \hat{x}}{\partial x} = \hat{S} K^T S_e^{-1} K \tag{3}$$

The diagonal elements of $A$ are the state vector elements' degrees of freedom, which provide a measure of the geophysical information obtained from the measurement through the OE process. Degrees of freedom close to 1 highlight a high contribution of the measurement in the estimation of a given state vector parameter, whereas degrees of freedom close to 0 denote a low contribution of the measurement and a high contribution of the a priori information. Finally, $A$ also enables the computation of $X_{CO_2}$ averaging kernel that describes its vertical sensitivity (Connor et al., 2008). It shows the atmospheric layers to which the retrieval is the most sensitive to and it is essential to characterize and correctly exploit the retrieved $X_{CO_2}$.

### 3.2.2 Forward and inverse setups for performance evaluation

We use the 5AI inverse model (Dogniaux et al., 2021) that relies on 4A/OP radiative transfer model (Scott and Chédin, 1981) to build the Jacobian matrix $K$. These forward radiative transfer simulations rely on GEISA 2015 spectroscopic database (Jacquinet-Husson et al., 2016) with line-mixing effects for $CO_2$ (Lamouroux et al., 2015) and collision induced absorption in the $O_2$ 0.76 µm band (Tran and Hartmann, 2008). Multiple scattering is taken into account through 4A/OP coupling with LIDORT (Spurr, 2002) and the aerosol optical properties are taken from the OPAC library, which uses lognormal size distributions (Hess et al., 1998). For the performance study performed here, we assume that these optical properties are perfectly known, including their spectral dependence, thus allowing the transfer of information when combining different spectral bands. Finally, the atmospheric model is discretized in 20 atmospheric layers that bound 19 layers, as done by the ACOS algorithm (O'Dell et al., 2018). Airglow emission which impacts MicroCarb 1.27 µm band (Bertaux et al., 2020) is not included here in 4A/OP simulations.

We include in the state vector all the main geophysical variables necessary to model SWIR spaceborne measurements. Those are listed in Table 4, along with their a priori values and uncertainties. This selection of state vector parameters is used for all exactly-defined concepts (with a small exception on albedo slope for NanoCarb) and CVAR experiment cases because the goal of this study is to explore the impact of design parameters on performance. Using a similar estimation scheme across all resolving power helps us achieve this goal. However, we do realise that, in practice, less geophysical elements are fitted for low resolving power observations (e.g. Cusworth et al., 2021). Finally, the measurement noise model that fills the diagonal of $S_e$ is calculated, as in Buchwitz et al. (2013), with a reference radiance $L_{\text{ref}}$ and a reference signal-to-noise ratio $SNR_{\text{ref}}$:

$$\sigma_e = \begin{cases} L_{\text{ref}}/SNR_{\text{ref}}, & \text{if } L < L_{\text{ref}} \\ L/\left(SNR_{\text{ref}}\sqrt{L/L_{\text{ref}}}\right), & \text{if } L \geq L_{\text{ref}} \end{cases} \tag{4}$$

with $\sigma_e$, the noise model for a given spectral sample, and $L$, the radiance for a given spectral sample. Here, the $S_e$ matrix only includes measurement noise, so the uncertainty (or precision) results obtained from the $\hat{S}$ matrix will only be related to measurement noise. In practice smoothing errors from $CO_2$ and non-$CO_2$ state vector parameters add up to the uncertainty (Connor et al., 2008). Finally, uncertainty evaluated on real data is typically larger than the uncertainty obtained from Optimal Estimation calculations, as model-related errors (uncertainty in albedo spectral dependence, spectroscopy errors, etc.) are also

encompassed in such evaluations. For example, at low resolving powers, the complexity of spectral dependence in surface reflectance can lead to significant errors (e.g. for methane, Ayasse et al., 2018) that will not be accounted for here or, at high resolving power, real-data uncertainties are evaluated to be twice as large as theoretical uncertainty for OCO-2 (Eldering et al., 2017).

**Table 4. State vector used for performance evaluation**

| Variable name | Length | A priori value | A priori uncertainty (1σ) | Notes |
|---|---|---|---|---|
| **H₂O scaling factor** | 1 | 1.0 | 0.5 | - |
| **CO₂ profile** | 19 layers | 394.95 ppm | Same matrix as ACOS (O'Dell et al., 2018) | - |
| **Surface Pressure** | 1 | 1013.0 hPa | 4.0 hPa | - |
| **Temperature Profile Shift** | 1 | 0 K | 5 K | - |
| **Surface albedo (order 0 of albedo model)** | 1-4 bands | true synthetic value | 1.0 | 4 bands in MicroCarb B1234 and NanoCarb cases |
| **Surface albedo (order 1 of albedo model)** | 1-4 bands | true synthetic value | 1.0 | 4 bands in MicroCarb B1234. Not included in the state vector for NanoCarb case, see Sect. 2.3. |
| **Coarse mode aerosol Optical Depth (COD)** | 1 layer | 0.02 | 0.1 | - |
| **Fine mode aerosol Optical Depth (COD)** | 1 layer | 0.05 | 0.1 | - |

## 4 Results and discussion for CVAR

### 4.1 Impact of spectral resolution and signal-to-noise ratio

This subsection explores the combined impact of spectral resolution and signal-to-noise ratio on $X_{CO_2}$ retrieval performance. First, we discuss how $X_{CO_2}$ precision and $CO_2$-related degrees of freedom evolve with spectral resolution and signal-to-noise ratio, and then we examine $X_{CO_2}$ vertical sensitivities.

### 4.1.1 $X_{CO_2}$ precision and degrees of freedom

Here, we assess the impact of varying spectral resolution and signal-to-noise ratio. For the atmospheric situation VEG-50º, Figure 3 shows the $X_{CO_2}$ precision (or random error) and degrees of freedom (hereafter DOFs) as a function of both the resolving power $\lambda/\Delta\lambda$ and the signal-to-noise ratio (SNR) for CVAR, and for the exact OCO-2, CO2M, MicroCarb and NanoCarb concepts (results for exactly-defined concepts are discussed in Sect. 5). The random error is computed from the a posteriori covariance matrix $\hat{S}$ given in Eq. (2), and the DOFs correspond to the sum of $CO_2$-related diagonal elements of matrix $A$, given in Eq. (3). As results change in values but conclusions do not for other albedo models and SZAs, figures that include all 12 atmospheric situations are shown in the Supplements.

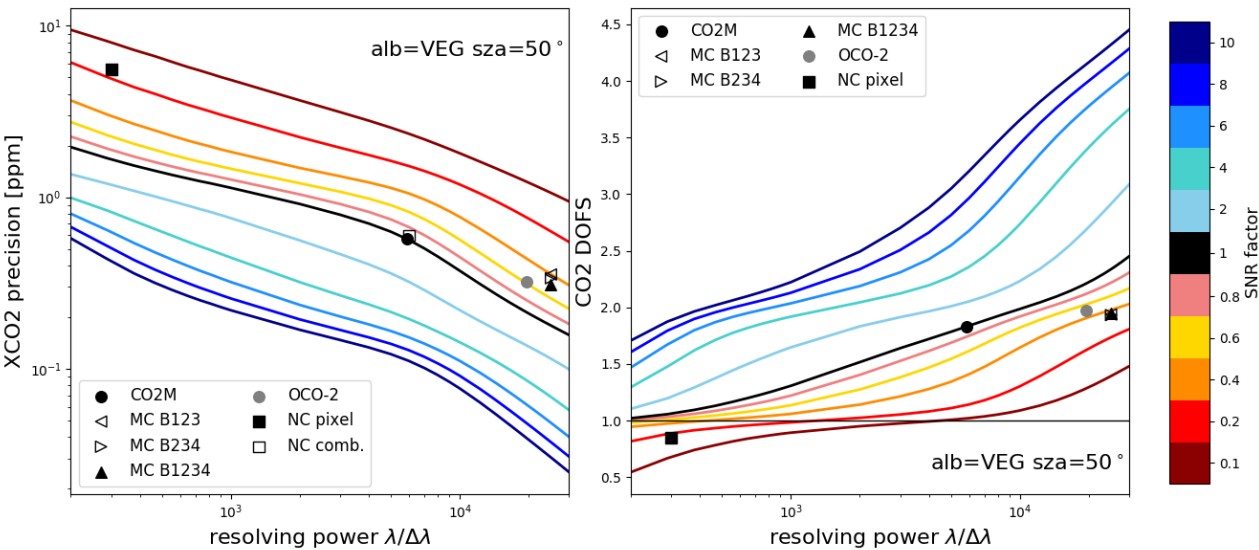

**Figure 3.** $X_{CO_2}$ **precision (left) and corresponding degrees of freedom for $CO_2$ (right), for the fictitious CVAR instrument for resolving power $\lambda/\Delta\lambda$ evolving from 200 to 30000 (horizontal axis), and for SNR evolving from 0.1 to 10 times CO2M reference SNR (colour scale), for the situation VEG-50º. Symbols give the same quantities for NanoCarb (NC, squares), MicroCarb (MC, triangles for various band combinations), CO2M (circle) and OCO-2 (grey circle). It should be noted that NanoCarb does not have a spectral resolution per se, the resolving powers used to plot its performance have been solely chosen for the sake of comparing NanoCarb and CVAR performance.**

Results for the reference CO2M SNR are given by the black line. $X_{CO_2}$ precision evolves from 1.96 ppm for $\lambda/\Delta\lambda$=200 to 0.16 ppm for $\lambda/\Delta\lambda$=30000. These values are consistent with those reported by previous studies: Galli et al. (2014) showed that degrading spectral resolution increased $X_{CO_2}$ random errors (values cannot be compared because no real measurement is processed here) and Wu et al. (2020) reported an increase of mean $X_{CO_2}$ retrieval noise from 0.25 ppm to 0.59 ppm (0.21 to 0.56 ppm in this work, respectively), when degrading OCO-2 measurements ($\lambda/\Delta\lambda \sim 20000$) to CO2M-like resolving powers ($\lambda/\Delta\lambda \sim 6000$). This improvement in precision with increasing resolving power is correlated to DOFs values that also increase with resolving power, from 1.02 to 2.45, respectively. Indeed, the more information a measurement can bring, the lower the

$X_{CO_2}$ random error. Changing the SNR has similar effects: the less noisy a measurement is, the more information it can carry, thus increasing SNR increases DOFs and reduces $X_{CO_2}$ random error. For example, for $\lambda/\Delta\lambda$=6000 (close to CO2M resolution), $X_{CO_2}$ precision evolves from 2.34 ppm to 0.11 ppm when multiplying SNR by 100. Overall, increasing the SNR by 2 orders of magnitudes improves the $X_{CO_2}$ precision by a factor ranging from 16 to 37 (for increasing resolving powers), whereas increasing the resolving power by 2.2 orders of magnitude (from $\lambda/\Delta\lambda$=200 to $\lambda/\Delta\lambda$=30000) only improves $X_{CO_2}$ precision

by a factor ranging from 10 to 23 (for increasing SNR values). Hence, it appears that $X_{CO_2}$ precision is more sensitive to SNR improvements rather than to resolving power improvements, for large improvements of two orders of magnitude centred on CO2M instrument characteristics. However, as it can be seen in Fig. 3 (and in Supplementary Fig. S3), this conclusion does not hold for smaller local improvements, that generally result in better $X_{CO_2}$ precision gains through resolving power improvements than through SNR improvements (see Supplementary Fig. S3).


Furthermore, $X_{CO_2}$ precision and DOFs grossly show two slope changes (in logarithmic scale) as the resolving power $\lambda/\Delta\lambda$ increases. Depending on SNR, the first one occurs around $\lambda/\Delta\lambda \sim 400 - 1000$. It corresponds to the complete P-R spectral band structure becoming visible, as previously commented for Fig. 2. Then, the second slope change occurs around $\lambda/\Delta\lambda \sim$ 4000 – 10000, it corresponds to the individual spectral lines of CO2 becoming clearly visible in spectral band branches, as also

commented for Fig. 2. Between these two slope changes ($\lambda/\Delta\lambda \sim 1000 - 4000$), improvements in resolving power are less efficient in improving $X_{CO_2}$ precision than elsewhere along the resolving power dimension (see Supplementary Fig. S3). This explains why, across the large two order of magnitude improvements in resolving power and SNR explored in this study, SNR has a larger impact on precision than resolving power. This result also underlines the critical importance of resolving new spectral features (the P-R band structure below $\lambda/\Delta\lambda \sim 1000$ or the individual spectral lines above $\lambda/\Delta\lambda \sim 4000$) to gain $X_{CO_2}$

precision efficiently.

In addition, we can note that these slope changes do not exactly occur for the same resolving power for different SNR values, and are more or less sharp. Indeed, the observation of band structures or individual spectral lines must be significant with respect to noise level to be actually able to bring information and improve $X_{CO_2}$ precision. We note that these slope changes

occur for smaller resolving powers as SNR increases, and that DOFs and $X_{CO_2}$ precision results for high SNR and low resolving powers correspond to results for low SNR and high resolving powers. Thus, a broad symmetry appears in the impact of SNR and resolving power on $X_{CO_2}$ retrieval performance.

### 4.1.2 Vertical sensitivity: column averaging kernels

In addition to information content (given by DOFs and, symmetrically, by precision), the vertical sensitivity (or column

averaging kernels, hereafter denoted AKs) of $X_{CO_2}$ retrievals must be examined (see Sect. 3.2.1). Taking into account the

vertical sensitivity of total columns is especially important when exploiting local column enhancements of vertically-inhomogeneous concentration increases. Indeed, any deviation from unity in the vertical sensitivity wrongfully scales differences between the unknown truth and the prior into the retrieved column enhancement, thus calling for a posteriori corrections, as presented by Krings et al. (2011) and also included by Borchardt et al. (2021) for aircraft observation processing.

Figure 4 shows AKs for CVAR with a resolving power varying from $\lambda/\Delta\lambda = 200$ to 30000, and for three different scaling factors of $CO_2M$ noise model (top row) and, conversely, shows AKs for scaling factors of $CO_2M$ noise model varying from 0.1 to 10, and for three different resolving powers (bottom row). Figure 4 only includes results for the situation VEG-50º. Results for the other situations are shown in the Supplements. For low SNR and resolving power values, AKs reach their maximum in the atmospheric layer closest to the ground, and have near-zeros values at the top of the atmosphere. As SNR or

resolving power (or both) increase, sensitivities for layers close to the ground improve, and become higher than one. For noise levels and a resolving power of about 6000 and above, the vertical sensitivity values are close to 1 from the ground surface up to approximately 300 hPa, and then decrease. For SNR and resolving power values even higher, AKs converge towards 1 for all atmospheric layers. Thus, as for their impact on $X_{CO_2}$ precision or DOFs, the resolving power and noise level of an observing concept also have very similar impacts on AK shape for CVAR.

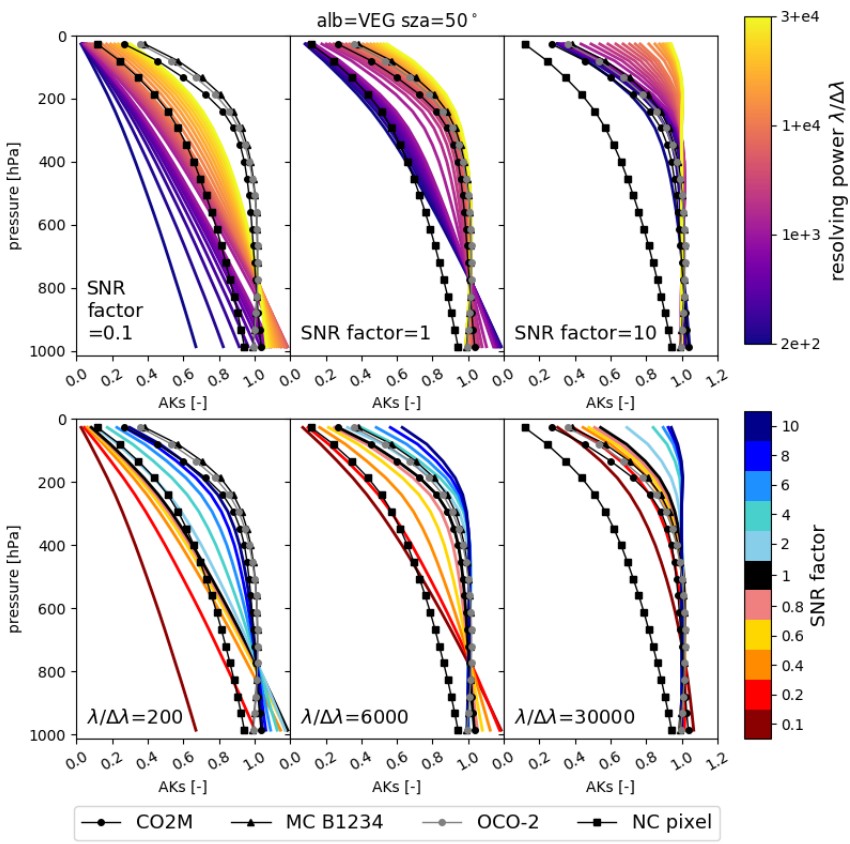


**Figure 4. Vertical sensitivities (AKs) as a function of resolving power $\lambda/\Delta\lambda$ for three different SNR scaling factors (top), and as a function of SNR for three different resolving powers $\lambda/\Delta\lambda$ values (bottom), for the observational situation VEG-50°. Black lines with symbols give vertical sensitivities for CO₂M (circles), MicroCarb (triangles), NanoCarb (squares) and OCO-2 (grey circles).**

### 4.2 Impact of spectral resolution and spectral band selection

This subsection explores the combined impact of spectral resolution and band selection on $X_{CO_2}$ retrieval performance. First, we discuss how $X_{CO_2}$ precision and $CO_2$ and non-$CO_2$ related degrees of freedom evolve with spectral resolution and band selection, and then we examine $X_{CO_2}$ vertical sensitivities. Finally, we explore $X_{CO_2}$ sensitivities to a priori misknowledge of interfering geophysical variables, with an eventual focus on aerosol-related parameters.

### 4.2.1 $X_{CO_2}$ precision and degrees of freedom for CO₂ and interfering geophysical variables

Here, we assess the impact of varying the spectral resolution and band selection. For the atmospheric situation VEG-50° (results for other situations are given in the Supplements), Figure 5 shows the $X_{CO_2}$ precision and DOFs as a function of both the

resolving power $\lambda/\Delta\lambda$ and spectral band selection for CVAR (with SNR fixed at its reference value), and for the exact OCO-2, CO2M, MicroCarb and NanoCarb concepts (results for exactly-defined concepts are discussed in Sect. 5). We can first note that including the $O_2$ 0.76 μm band (denoted B1 in Fig. 5) increases $CO_2$ DOFs for CVAR cases, compared to cases where it is not included (denoted B2, B3 and B23 in Fig. 5). This spectral band is indeed sensitive to surface pressure, temperature and aerosols, and can thus bring independent constraint on these geophysical parameters, that show sensitivities that correlate with $CO_2$ sensitivity in 1.6 and 2.05 μm bands. For resolving powers above 1000, adding the $O_2$ 0.76 μm band has less impact on $X_{CO_2}$ precision for B2 than for B3 cases. This may be explained by the fact that spectral lines are more saturated in B3, thus providing less information regarding the length of the optical path than in B2. Besides, we can also notice that $CO_2$ DOFs for B3/B13 are always higher than for B2/B12 cases. This may be explained by the fact that CO2M 2.05 μm band includes two full sets of $CO_2$ P-R absorption branches (out of the three present near 2.05 μm, with one more saturated than the other, see Fig. 1), whereas there is only one set of $CO_2$ branches in B2 near 1.6 μm, for identical SNR values between B2 and B3. Thus, B3 carries more $CO_2$ information than B2. Interestingly, we can also notice that band configurations with the higher DOFs do not systematically translate into better $X_{CO_2}$ precision: for example B3/B13 always shows higher DOFs than B2/B12, but very similar $X_{CO_2}$ precisions from $\lambda/\Delta\lambda$=3000 and upwards. This is due to the covariance between $CO_2$ elements in the state vector that vary between band selection cases (see Supplementary Figure S18), which shows that different spectral bands carry different $CO_2$ information.

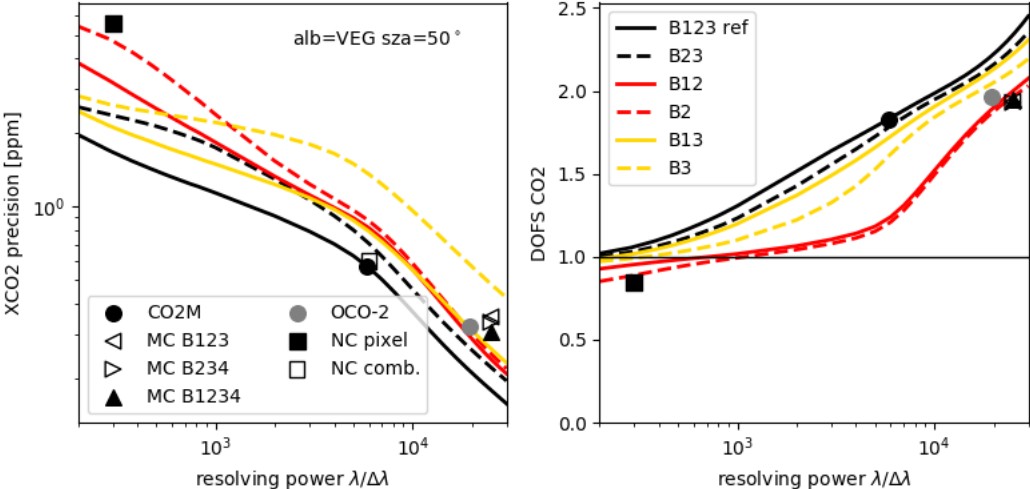

**Figure 5.** $X_{CO_2}$ **precision (left), and corresponding degrees of freedom for CO₂ (right), for the fictitious CVAR instrument for resolving power $\lambda/\Delta\lambda$ evolving from 200 to 30000 (horizontal axis), and for different spectral band selections: with and without O₂ 0.76 μm band (B1, full and dashed-lines, respectively), with both CO₂ 1.6 and 2.05 μm bands (B23, black), with only the 1.6 μm band (B2, red) and with only the 2.05 μm band (B3, yellow), for the situation VEG-50°. Symbols give the same quantities for NanoCarb (NC, squares), MicroCarb (MC, triangles for various band combinations), CO₂M (circle) and OCO-2 (grey circle). It should be noted that NanoCarb does not have a spectral resolution per se, the resolving powers used to plot its performance have been solely chosen for the sake of comparing NanoCarb and CVAR performance.**

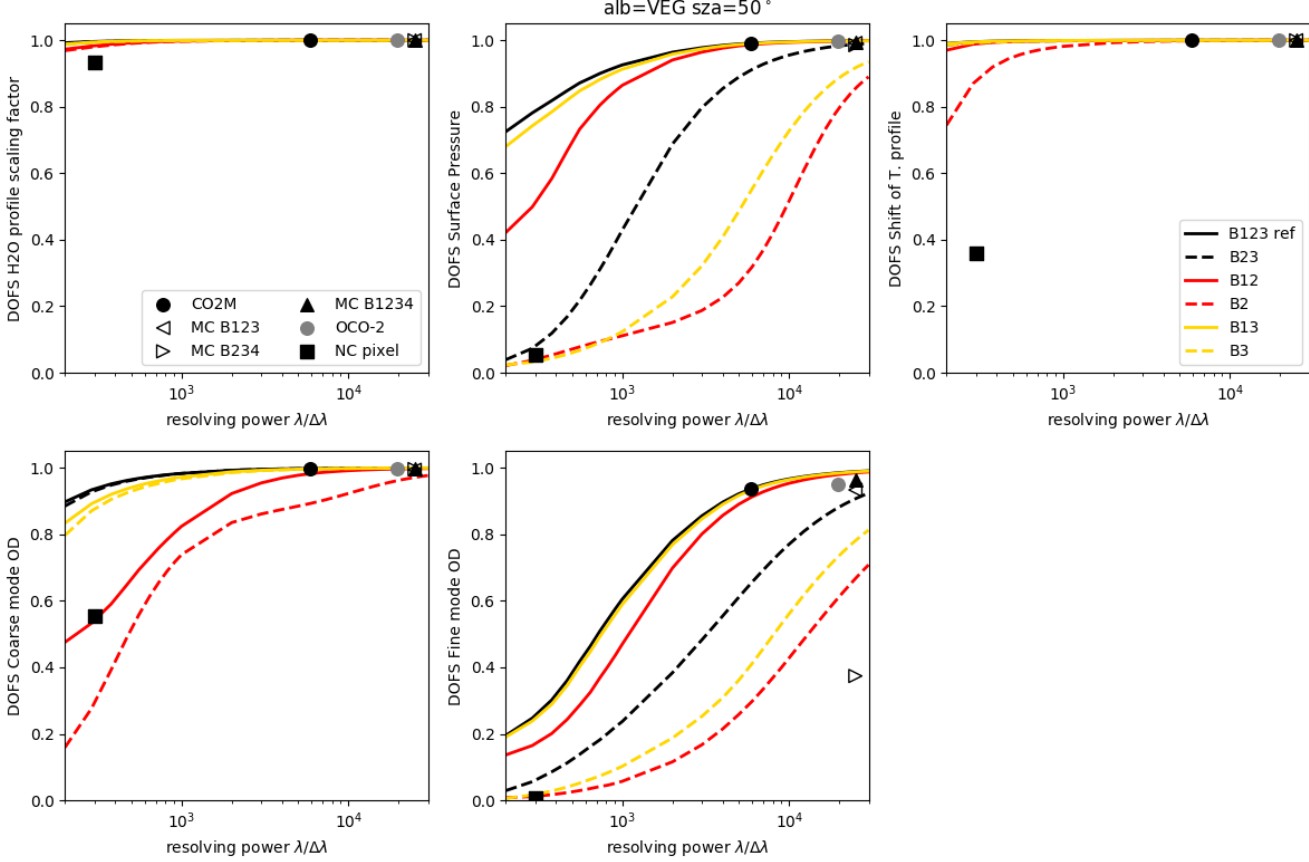

**Figure 6. Degrees of freedom for H2O scaling factor (top, left), Surface pressure (top, center), Temperature profile shift (top, right), Coarse mode aerosol optical depth (bottom, left) and Fine mode aerosol optical depth (bottom, center), for the fictitious CVAR instrument for resolving power $\lambda/\Delta\lambda$ evolving from 200 to 30000 (horizontal axis), and for different spectral band selections: with and without $O_2$ 0.76 µm band (B1, full and dashed-lines, respectively), with both $CO_2$ 1.6 and 2.05 µm bands (B23, black), with only the 1.6 µm band (B2, red) and with only the 2.05 µm band (B3, yellow), for the situation VEG-50º. Symbols give the same quantities for NanoCarb (NC, squares), MicroCarb (MC, triangles for various band combinations), $CO_2M$ (circle) and OCO-2 (grey circle). It should be noted that NanoCarb does not have a spectral resolution per se, the resolving powers used to plot its performance have been solely chosen for the sake of comparing NanoCarb and CVAR performance.**

For the atmospheric situation VEG-50º (results for other situations are given in the Supplements), Figure 6 completes Fig. 5 by showing DOFs for interfering geophysical variables (H2O profile scaling factor, surface pressure, temperature profile shift and aerosol optical depths; albedo-related parameters are not included because they are all very close or equal to 1) as a function of both the resolving power $\lambda/\Delta\lambda$ and spectral band selection for CVAR (with SNR fixed at its reference value), and for the exact OCO-2, $CO_2M$, MicroCarb and NanoCarb concepts (see Sect. 5). For all five variables, DOFs increase with resolving powers and tend towards 1. H2O profile scaling factor and temperature profile shift exhibit high close-to-one DOFs for almost all resolving powers and spectral band selection cases in the configuration used here. Surface pressure and aerosol optical depths, variables that influence the length of the optical path, have more sensitivity to resolving power, and also especially to

the inclusion – or not – of the $O_2$ 0.76 μm band (B1) in the spectral band selection. Cases that do not include B1 show much lower DOFs for these variables, illustrating once again, how useful this band is to constrain interfering geophysical variables. This result is made possible by the usual (see OCO-2 processing algorithm ACOS for example, O'Dell et al., 2018) hypothesis of fixed aerosol optical properties, which enables sharing optical path information across spectral bands. Overall, we can also note that, for all geophysical variables, DOFs for B13 are more or less significantly closer to those of B123, compared to DOFs

of B12. This shows that the $CO_2$ 1.6 μm band only brings little complementary interfering variable information on the top of the one already carried by the 2.05 μm band.

Previous studies that have explored the impact of spectral band selection and/or spectral resolution on $X_{CO_2}$ performance provide conclusions in broad agreement with the previously presented results. Wilzewski et al. (2020) studied the performance

of $X_{CO_2}$ retrievals from spectrally-degraded GOSAT measurements only using the 1.6 or 2.05 μm spectral bands. While methodologies are hardly comparable (because this study is only based on synthetic simulations), both works agree that the $X_{CO_2}$ precision and resolving power relationship has a change of characteristic around $\lambda/\Delta\lambda = 1000 - 2000$, when solely using the 1.6 or 2.05 μm $CO_2$ bands (see Supplementary Fig. S11 for Fig. 5 plotted in linear scale). Building on Wilzewski et al. (2020), Strandgren et al. (2020) select the 2.05 μm $CO_2$ band for the design of a moderate resolution instrument, partly because

it shows scattering particle sensitivity. Results presented here are consistent with this conclusion: using the sole 2.05 μm band yields higher (or equal for surface pressure at low resolving powers) DOFs for all geophysical variables, compared to using the sole 1.6 μm $CO_2$ band.

### 4.2.2 Vertical sensitivity: column averaging kernels

Figure 7 gives the column averaging kernel – which describes $X_{CO_2}$ vertical sensitivity – for all CVAR spectral band selection

cases, and for three different resolving power values (200, 6000 and 30000). For lower resolving powers, spectral band selection cases that include the $CO_2$ 2.05 μm band (B3) show a greater sensitivity to atmospheric levels close to the surface. This may be explained by the fact that this spectral band includes saturated spectral lines which are more sensitive to $CO_2$ concentration variations in atmospheric layers close to the surface, as it can be seen for example in Fig. 2 in Roche et al. (2021). As for Fig. 4, AKs tend to converge towards unity when resolving power increases, and this difference between bands

disappears. Besides, it can be noted that, comparing AKs between B23 and B123 spectral band selection cases, including or not an $O_2$ sensitive band does not have a strong impact on $X_{CO_2}$ vertical sensitivity.

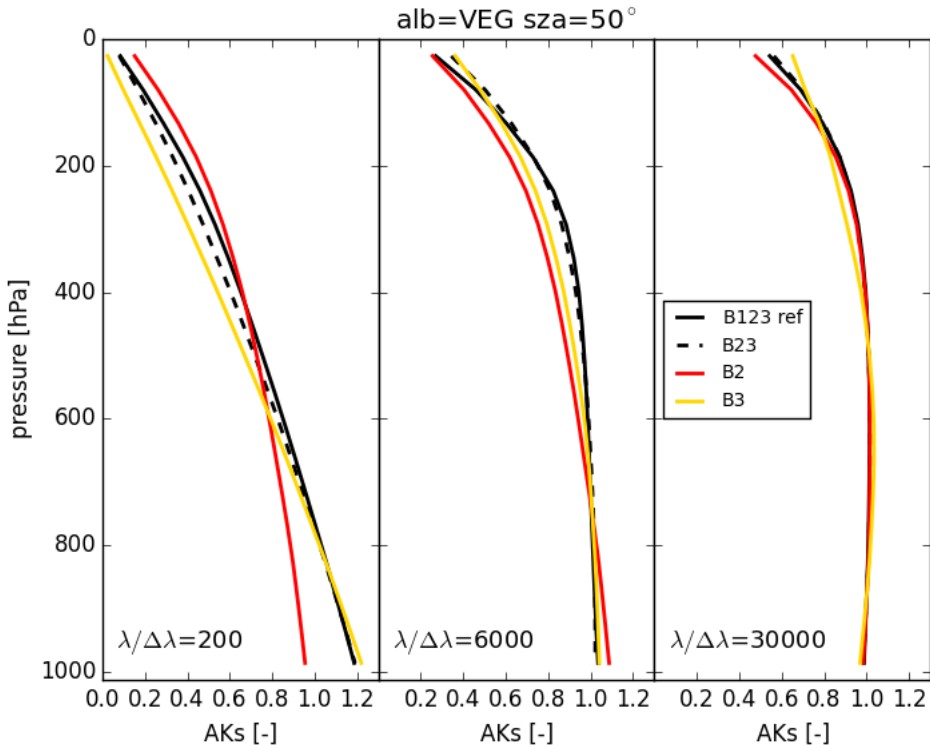

**Figure 7. Vertical sensitivities (AKs) for different spectral band selections (colours and line styles, see legend), and three different resolving powers $\lambda/\Delta\lambda$ values, for the observational situation VEG-50°.**

**4.2.3 Geophysical information entanglements**

Geophysical information is more or less entangled in SWIR measurements (as illustrated for example by a posteriori correlation matrices shown in the Supplements S18 or in Fig. 10), depending on the measurement nature (spectra or truncated interferogram), and also on its characteristics (spectral band selection, spectral resolution, SNR, etc.). One consequence of these entanglements is that possible a priori misknowledge of the atmospheric state can impact retrieved $X_{CO_2}$ and cause biases:

this is called smoothing error (Connor et al., 2008; Rodgers, 2000). In this section, we use the averaging kernel matrix $A$ to propagate a priori misknowledge of the synthetic true state of the atmosphere for non-$CO_2$ interfering variables in order to evaluate its impact on retrieved $X_{CO_2}$. Figure 8 shows, for the 12 atmospheric and observational situations considered in this work, the $X_{CO_2}$ impact of a priori misknowledge for several state vector variables for all CVAR spectral band selection cases, for resolving power values ranging from 200 to 30000, as well as for the exact MicroCarb, CO2M and NanoCarb concepts (see

Sect. 5 for exactly-defined concepts). These a priori perturbations include: (1) +10% to $H_2O$ profile scaling factor; (2) + 1 hPa to surface pressure; (3) + 1 K to the temperature profile shift; and (4) +0.05 to the albedo in 1.6 µm band. All situations and

perturbations are sorted along a unique axis, and the top three panels in Fig. 8 describe both the situations (albedo and SZA) and perturbations considered.

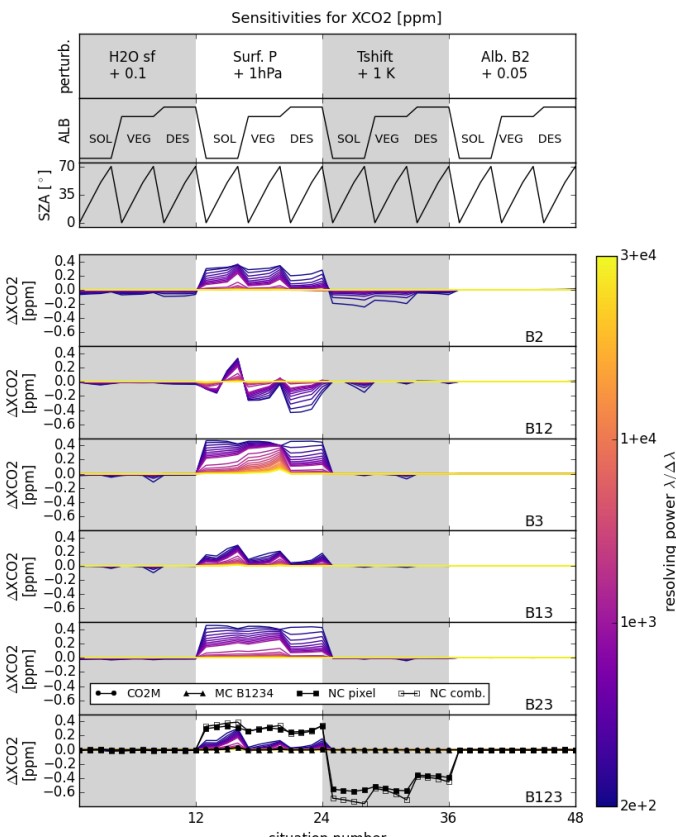

Figure 8. $X_{CO_2}$ sensitivities (noted $\Delta X_{CO_2}$) to prior misknowledge of water vapor, surface pressure, temperature profile shift and 1.6 µm band albedo value (described in the top panel), for 12 observational situations (described in the 2nd and 3rd panels), for 6 different CVAR spectral band selections (lines in the 6 bottom panels) and resolving power values ranging from 200 to 30000 (color scale). Black lines with symbols give the same sensitivities for $CO_2$M (circles), MicroCarb (triangles) and NanoCarb (squares) in the bottom panel.

First, regarding water vapour, $X_{CO_2}$ sensitivities are very small for all CVAR band selection configurations (consistently with results in Fig. 6), spectral resolutions and exactly described concept: for VEG-70° in CVAR B3 case, for the lowest resolving power, it amounts to a maximum of 0.12 ppm, in absolute value. This means that if a water vapor plume is correlated with a $CO_2$ emission plume, in the exhaust fumes of a coal-fired power plant for instance, a small bias in retrieved $X_{CO_2}$ enhancement could then hamper estimations from a low resolving power (but potentially high spatial resolution) instrument. However,

considering that emission rates computed from enhancements with mass-balance approaches may have uncertainties up to 65% (mainly due to wind-speed errors, Varon et al., 2018), this sensitivity of retrieved $X_{CO_2}$ to water vapour appears insignificant.

Regarding surface pressure, $X_{CO_2}$ sensitivities are close to zero for all resolving powers above 10000, and increase differently depending on spectral band selection case for lower resolving powers. They reach up to 0.47 ppm for CVAR B3 case at the lowest resolving power value tested. Overall, $X_{CO_2}$ sensitivities to prior surface pressure misknowledge is reduced when the $O_2$ 0.76 µm band is included in the measurement, consistently with results shown in Fig. 6. These sensitivities can be expected to impact the full-swath of an imaging instrument with lower resolving powers, and thus can be removed when computing an enhancement. However, they would blindly impact observations without emissions plumes to detect, thus making these observations hard to exploit for other purposes than anthropogenic point source monitoring.

Regarding temperature global shift, consistently with high DOFs shown in Fig. 6, $X_{CO_2}$ sensitivities are overall very small for spectral resolving power above 1000 and all CVAR spectral band selection cases. For resolving powers lower than 1000, they can reach up to 0.23 ppm in absolute value, for the CVAR B2 case for instance (see lower DOFs in Fig. 7).

Finally, all cases and concepts exhibit near-zero (or even, by construction, exactly-zero for B3 and B13 CVAR cases) $X_{CO_2}$ sensitivities when perturbating the 1.6 µm $CO_2$ band albedo by 0.05. This reflects the albedo DOFs very close or equal to 1 that we obtain with this inverse setup configuration, as well as the low posterior correlations between albedo and $CO_2$ parameters in the state vector.

### 4.2.4 Focus on sensitivities to prior aerosol misknowledge

We follow the approach used for NanoCarb performance assessement (Dogniaux et al., 2022), that was first introduced by Buchwitz et al. (2013) for the performance assessment of CarbonSat. Considering the previously described 12 atmospheric and observational situations that span three different albedo models and four SZA values, we explore the $X_{CO_2}$ sensitivities for synthetic coarse mode aerosol optical depths spanning 0.001 – 0.15 (with a fixed prior of 0.02) and fine mode aerosol optical depths spanning 0.001 – 0.22 (with a fixed prior of 0.05), thus yielding 192 situations in total. This a priori misknowledge of aerosol optical depths is propagated through the averaging kernel matrix $A$ to evaluate its impact on retrieved $X_{CO_2}$ values.

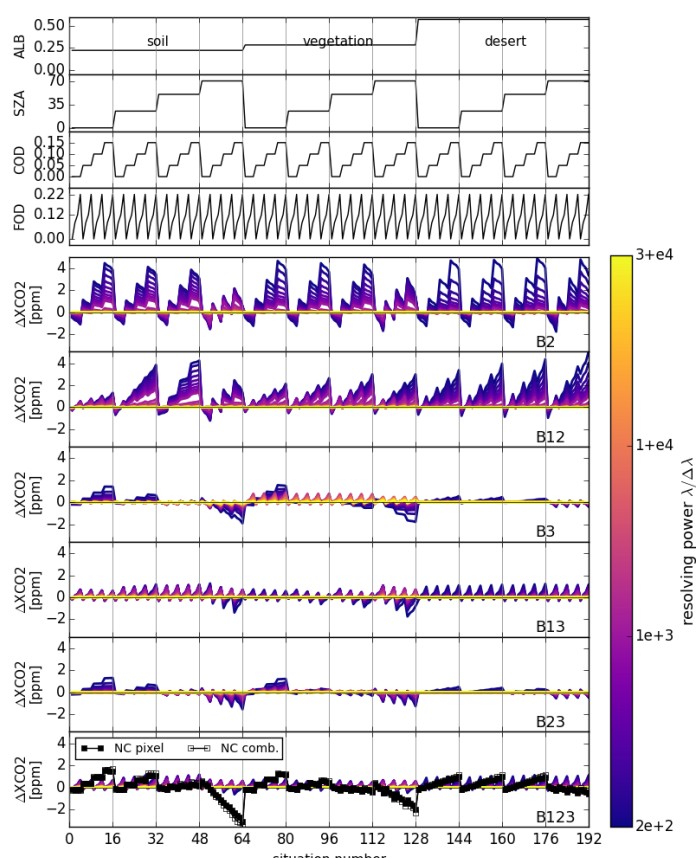

**Figure 9.** $X_{CO_2}$ sensitivities (noted $\Delta X_{CO_2}$) to prior misknowledge of aerosol optical depths (described in the 3$^{rd}$ and 4$^{th}$ top panels), for 12 observational situations (described in the 1$^{st}$ and 2$^{nd}$ panels), for 6 different CVAR spectral band selections (lines in the 6 bottom panels) and resolving power values ranging from 200 to 30000 (color scale). Black lines with symbols give the same sensitivities for NanoCarb (squares) in the bottom panel.

Figure 9 shows for the 192 considered situations, the $X_{CO_2}$ impact of a priori aerosol optical depth misknowledge for all CVAR spectral band selection cases, for resolving power values ranging from 200 to 30000, as well as for the exact NanoCarb concept (results for NanoCarb are discussed in Sect. 5). All situations are sorted along a unique axis, and the top four panels in Fig. 9 describe a given situation (albedo and SZA, coarse and fine mode optical depths).

For CVAR B2 and B12 cases, $X_{CO_2}$ sensitivities at low resolving powers reach up to about ~5 ppm. They mostly correlate with a priori misknowledge of coarse mode aerosol optical depth, and secondarily to fine mode aerosol optical depth, and diminish

as spectral resolving power increases. Sensitivities are also different depending on albedo model and SZA values, thus reflecting that the information content carried by a given measurement also depends on the scene (see DOFs for all situations in the Supplements). Including the $O_2$ 0.76 µm band in addition to the $CO_2$ 1.6 µm band reduces $X_{CO_2}$ sensitivities to a priori aerosol misknowledge at low SZA values for low resolving powers, but show low or even detrimental impacts at higher SZAs.

Unlike $CO_2$ 1.6 µm band, the $CO_2$ 2.05 µm band carries more aerosol information and thus results for CVAR B3 and B13 selection cases show lower impacts of these variables on $X_{CO_2}$ retrievals, with a maximum of ~2 ppm in absolute value. For CVAR B3 case, $X_{CO_2}$ sensitivities to a priori aerosol misknowledge are mostly correlated to coarse mode misknowledge for low resolving powers and to fine mode misknowledge for higher resolving powers, converging towards near-zero values for the highest resolving powers. Interestingly, including the $O_2$ 0.76 µm band changes this correlation pattern and $X_{CO_2}$ sensitivities appear to be mostly correlated to fine mode aerosol misknowledge for B13 case, with slightly higher $X_{CO_2}$ sensitivities compared to B3 case for some situations, such as those with soil or desert-like albedo.

Results for B23 and B123 mostly follow the patterns showed by B3 and B13 cases (as B3 brings most of the aerosol information compared to B2), only with slightly lower $X_{CO_2}$ sensitivity values, reflecting the little complementary information brought by B2 on the top of B3.

## 5 Results and discussion for exactly-defined concepts: OCO-2, CO₂M, MicroCarb and NanoCarb

## 5.1 $X_{CO_2}$ precision and degrees of freedom

Besides results for CVAR, Fig. 3 also includes the performance computed for four explicit concepts: the currently-flying OCO-2 and upcoming CO₂M and MicroCarb missions, as well as the NanoCarb concept that is currently being studied. First, OCO-2 shows a noise-only related precision of 0.32 ppm corresponding to DOFs for $CO_2$-related parameters of 1.97. The OCO-2 results that we obtain are overall consistent with ACOS results for soundings with close band-wise albedo values (see Supplementary Figure S6). Besides, land nadir OCO-2 $X_{CO_2}$ retrievals show an overall 0.77 ppm standard deviation compared to the Total Carbon Column Observing Network (TCCON) validation reference (Taylor et al., 2023). This difference with respect to the theoretical uncertainty computed from Optimal Estimation stems from all the forward and inverse modelling errors that are not accounted for in the retrieval scheme. Thus, this illustrates that the results provided in this study are a lower bound to the actual precisions that these upcoming concepts will have. CO₂M shows an $X_{CO_2}$ precision of 0.56 ppm, which is consistent with the 0.7 ppm precision requirement for a vegetation scene with SZA=50º given in (Meijer, 2020). For MicroCarb (MC1234, when including the 4 spectral bands), we find an $X_{CO_2}$ precision of 0.31 ppm that satisfactorily compares to the median 0.35 ppm contribution of SNR to the mission error budget (with the full range of possible contribution being 0.15 – 0.94 ppm, personal communication). Besides, we can notice that removing one of the two $O_2$-sensitive spectral band from

MicroCarb measurement slightly decreases precision (MC123 for 0.76-1.6-2.05 μm bands, and MC234 for 1.6-2.05-1.27 μm bands). Indeed, less geophysical information is available to help constrain interfering variables. Finally, MicroCarb (MC1234) shows only slightly higher $CO_2$ DOFs compared to CO2M despite having a spectral resolution 5 times higher: this may be explained by the fact that their respective spectral bands are not covering the same wavelength intervals, as it can be seen in Fig. 1.

Two different $X_{CO_2}$ precision results are included for NanoCarb in Fig 3.: one given for a unique pixel located at the FOV centre (filled square), and another one obtained after combining results acquired from different viewing angles as the two-dimensional FOV of NanoCarb flies over a scene (see Sect 2.3, or the extensive description in Dogniaux et al., 2022). For a unique observation of a given scene, performed by the FOV central pixel, NanoCarb yields a precision of 5.6 ppm. However, after combining the maximum 102 observations (over different viewing angles) of the same scene, NanoCarb random error is reduced to 0.60 ppm, which is close to CO2M performance. It must be noted that, because of its very nature, NanoCarb does not have a spectral resolution per se. Thus, we arbitrarily attributed a resolving power of $\lambda/\Delta\lambda = 300$ to plot NanoCarb pixel-wise performance, and a resolving power $\lambda/\Delta\lambda = 6000$ to plot NanoCarb performance for combined pixels. This choice enables to highlight that NanoCarb pixel-wise performance compares to concepts measuring spectra at low spectral resolution and low SNR, whereas once retrieval results from observations that are assumed independent are combined, the $X_{CO_2}$ precision compares to CO2M.

However, despite similar precisions, further comparisons enable to exhibit how their respective $X_{CO_2}$ observations are not equivalent. Indeed, we can first notice that their respective DOFs are not comparable at all: CO2M shows 1.83 DOFs for $CO_2$, whereas NanoCarb pixel-wise $CO_2$ DOFs amount to 0.85 (no averaging kernel matrix is computed for NanoCarb combined-pixel results, as performance is evaluated per pixel and the combination then assumes that they are independent). These low $CO_2$ DOFs for NanoCarb are explained by the low $CO_2$ information content of NanoCarb measurement (compared to concepts that measure spectra), and its entanglement with other geophysical variable information, as mentioned in Dogniaux et al. (2022). One can indeed notice that $CO_2$ and other variables' partial derivatives are correlated (see Fig S1 in the Supplements): this makes it harder for OE to yield independent estimates for $CO_2$ and other parameters.

Another way to look at this issue is to consider a posteriori correlations between state vector parameters, as given by the a posteriori covariance matrix $\hat{S}$. Pearson correlation coefficient matrices computed from $\hat{S}$ are shown for CO2M, MicroCarb (MC1234) and NanoCarb in Fig. 10. First, regarding $CO_2$ profile, we notice the very high positive correlation between different atmospheric layers for NanoCarb, compared to CO2M and MicroCarb cases. This is a result of the low $CO_2$ information content in NanoCarb measurements, compared to CO2M and MicroCarb. Besides, in this state vector configuration, NanoCarb a posteriori covariance matrix shows stronger correlation between $CO_2$ atmospheric layers and temperature profile shift than for

CO2M and MicroCarb. A slight positive correlation between $CO_2$ atmospheric layers and albedo parameters can also be noted for NanoCarb, unlike CO2M and MicroCarb that show small negative correlations. The close-to-1.0 correlation between albedo parameters of different bands are due to the presence of aerosol optical depth parameters in the state vector (assuming fixed aerosol optical properties). When aerosol optical depths are removed from the state vector, these correlations between albedo parameters of different bands decrease (see Supplements S19). Interestingly, in that case, correlations between $CO_2$ atmospheric layers and albedo parameters reach up to 0.6 for NanoCarb, whereas they reach 0.2 and 0.1 for CO2M and MicroCarb, respectively (see Supplements). Thus, all things considered, NanoCarb contains less geophysical information than other concepts that measure spectra, and such a comparison helps pave the way for future improvement of the NanoCarb concept.

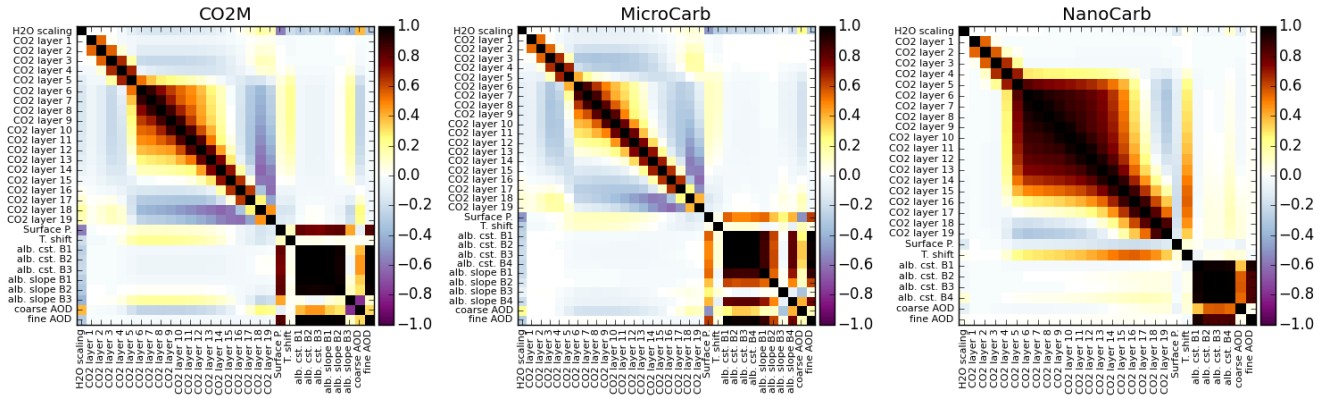

**Figure 10. A posteriori correlation matrices for CO2M (left), MicroCarb (center) and NanoCarb (right), for the VEG-50º situation.**

## 5.2 Vertical sensitivity: column averaging kernels

Besides results for CVAR, Fig. 4 also includes the vertical sensitivities of OCO-2, CO2M, MicroCarb and NanoCarb. Thanks to their spectral resolution and SNR, CO2M and MicroCarb show vertical sensitivities that are close to 1 from the surface to about 300 hPa, and that then decrease. Interestingly, NanoCarb AKs do not exactly share the shape of CVAR, CO2M or MicroCarb AKs. For the atmospheric layers closest to the ground, NanoCarb AK follows the AK shape of an instrument with low resolving power and SNR (consistently with how NanoCarb single-pixel performance compares to CVAR in Fig. 3). For higher up atmospheric layers, it follows the AK shape of an instrument with medium resolving power and low SNR, or vice versa.

## 5.3 Non-CO2 degrees of freedom

Besides results for CVAR, Fig. 6 also gives interfering variable DOFs for the exact OCO-2, CO2M, MicroCarb and NanoCarb concepts. For MicroCarb and OCO-2, CO2M, DOFs for $H_2O$ profile scaling factor, surface pressure, temperature global shift and coarse mode aerosol optical depth are all nearly equal to 1. Only for fine mode aerosol optical depth do DOFs appear to

be – slightly – lower than 1, with the exception of MicroCarb B234 configuration test, where fine mode DOF is close to 0.4. This shows that different optical path length information is carried depending on whether the $O_2$ 0.76 µm or 1.27 µm bands are used. NanoCarb exhibits near-zero DOFs for surface pressure and fine mode optical depth in this retrieval configuration, as well as non-zero yet rather low DOF values for $H_2O$ profile scaling factor, temperature global shift and coarse mode aerosol optical depth (0.93, 0.36 and 0.55, respectively). Given how little some of these DOF values are, we also conclude that the state vector used to process NanoCarb measurement should be adjusted to only include the most essential geophysical variables. For example, aerosol optical depths could be removed.

## 5.4 Geophysical information entanglements

Besides results for CVAR, Fig. 8 also gives CO2M, MicroCarb and NanoCarb $X_{CO_2}$ sensitivities to a priori misknowledge of water vapour, surface pressure, temperature profile, and 1.6 µm albedo. These $X_{CO_2}$ sensitivities are close to zero for MicroCarb and CO2M, for the four tested geophysical variables.

Prior misknowledge of surface pressure reach up to 0.39 ppm for NanoCarb, showing again results comparable to low spectral resolution instruments. Sensitivities for NanoCarb central pixel (filled squares) and NanoCarb combined results for the central row of pixels (empty squares) slightly differ because information entanglement evolves depending on the pixel location in the NanoCarb FOV. NanoCarb also shows significant $X_{CO_2}$ sensitivities to a priori temperature misknowledge that reach -0.76 ppm, consistently with the strong correlation shown between the temperature profile shift and $CO_2$ related parameters in the state vector (see Fig. 10). This result is not surprising as the version of the NanoCarb concept used in this work did not consider the possible impact of entanglements between $CO_2$ and temperature (see Sect 2.3 or Dogniaux et al., 2022, Gousset et al., 2019). This paves the way for future improvements of this very compact instrumental concept.

Regarding the sensitivity to prior aerosol misknowledge, Fig. 9 also shows results for the exact NanoCarb concept, for both the central FOV pixel and the combination of the central along-track row of pixels. For SZA values lower or equal to 50º (in soil and vegetation albedo situations) and for all SZAs in desert albedo situations, those mostly correlate with coarse mode aerosol misknowledge, and reach absolute values up to 1.7 ppm. For SZA=70º in soil and vegetation albedo situations, NanoCarb aerosol DOFs not only increase (see Fig. S16-17 in Supplements), but correlations between $CO_2$ and aerosol state variables also increase by a lot (see Fig. S20 in Supplements), thus leading to the larger sensitivities shown in Fig. 9 (or Fig. 11). This illustrates that different surface types and SZA must be explored for thorough performance assessments.

Figure 11 is similar to Fig. 9, but focuses on the exact concepts studied of CO2M and MicroCarb. Their $X_{CO_2}$ sensitivities to prior aerosol optical depth misknowledge is overly correlated to the one of fine mode optical depth. It can be explained by the fact that their coarse mode optical depth DOFs are very close to 1, thus enabling a correct estimation (in this synthetic

simulation set up) of this geophysical parameter, which results in a very low impact of coarse mode optical depth
misknowledge on $X_{CO_2}$ retrievals. However, their fine mode DOFs are below 1, thus leading to estimation errors that impact $X_{CO_2}$ retrievals through a posteriori correlations (see Fig. 10 or Supplements S19). Overall, CO2M shows sensitivities up to 0.2 ppm, with maximums reached for the SOL-25º situation. These values are well below the 0.5 ppm systematic error requirement (Meijer, 2020), and are expected to be even more reduced by using the aerosol observations provided by the Multi-Angle Polarimeter that will fly along CO2M spectrometers (Rusli et al., 2021). As for MicroCarb, its $X_{CO_2}$ sensitivities to
aerosol optical depth misknowledge measurement peak for soil-albedo situations up to about 0.6 ppm when just including the 0.76 μm $O_2$ band (B123), and up to 0.1 ppm when both 0.76 and 1.27 μm $O_2$ bands are included (B1234), whereas it peaks up to -0.2 ppm for vegetation-albedo situations when only the 1.27 μm $O_2$ band is available (B234). Interestingly, sensitivities for MicroCarb B123 and B1234 are positively correlated to fine mode aerosol optical depth values, whereas MicroCarb B234 are negatively correlated. This illustrates that the $O_2$ 1.27 μm band carries complementary optical path length information
compared to the $O_2$ 0.76 μm band (see also a posteriori correlation matrices in Supplements S21).

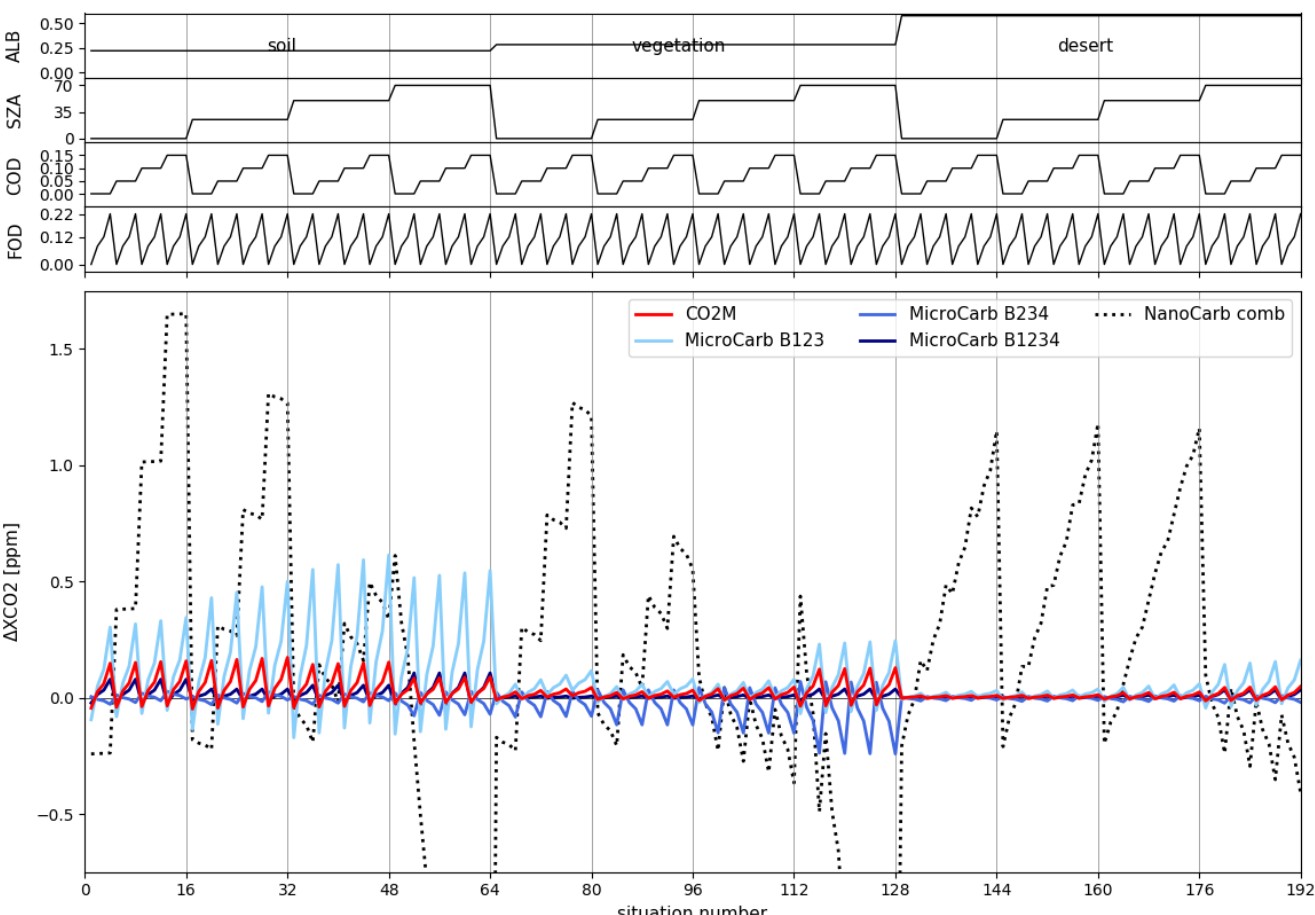

Figure 11. Same as Fig. 10, but containing only results for CO2M, MicroCarb and NanoCarb.

## 6 Conclusions

In this work, we have carried out a synthetic survey that describes the impact of measurement design choices on $X_{CO_2}$ retrieval
performance, for shortwave infrared (SWIR) satellite observations. In order to be representative of the large extent of upcoming
concept designs, it explored – for a fictitiously-varying CO2M-like instrument – the impact of three different parameters: (1)
spectral resolution; (2) signal-to-noise ratio, for values spanning two orders of magnitude, and (3) spectral band selection
within the measurement. In addition, four exactly-defined concepts have been consistently studied: CO2M, MicroCarb and
NanoCarb.

First, $X_{CO_2}$ precision and $CO_2$ information content of SWIR measurements improve when increasing either or both resolving
power and signal-to-noise ratio. For CO2M-like SNR values, increasing the resolving power by 2.2 orders of magnitude enables
to improve precision by a factor of about 12. For a resolving power of 6000, multiplying SNR by 100 enables to improve
precision by a factor of about 21. Overall, for these large changes of about two orders of magnitude, precision is more sensitive
to SNR improvement than to spectral resolution improvements. However, small magnitude improvements in resolving power
generally yield more $X_{CO_2}$ precision improvements than SNR improvements, especially when $CO_2$ spectral lines are resolved.
The separate impacts of these two parameters show a broad symmetry in precision, as well as in vertical sensitivities:
measurements with lower SNR and/or spectral resolution give more weight to atmospheric layers close to the surface for the
retrieved total columns.

The comparison of different spectral band selections included in a SWIR measurement provided two main conclusions. First,
including the $O_2$ 0.76 µm band strongly increases information content for parameters impacting the optical path length (for all
resolving powers), and helps to noticeably reduce the impact of a priori misknowledge of these parameters on retrieved $X_{CO_2}$
values. Secondly, the $CO_2$ 2.05 µm band (and especially for coarse aerosol mode) carries overall more information than the
1.6 µm band, and thus seems more appropriate for concepts that measure a single $CO_2$-sensitive spectral band, especially at
low to mid-spectral resolution values. These results also highlight how the precise (and accurate to some extent) retrieval of
$X_{CO_2}$ from SWIR observations relies on the amount of information carried by these observations. Reducing spectral resolution
and/or the number of spectral bands to improve spatial resolution increases errors. If those are constant over a full-image, they
may be removed when calculating local enhancements of $X_{CO_2}$. However, they may still hamper absolute $X_{CO_2}$ retrievals in
plume-free scenes, thus potentially making these observations hardly useful for anything but anthropogenic emission imaging.

The exact CO2M, MicroCarb and NanoCarb concepts were also studied in addition to the fictitiously-varying ones. With an
about 5-times higher spectral resolution but shorter spectral band intervals, MicroCarb shows slightly higher $CO_2$ DOFs
compared to CO2M, and its $X_{CO_2}$ precision is lower by a factor ranging from about 1.1 to 1.9, depending on observational
situations. Their vertical sensitivities are very similar, and their DOFs for other interfering variables are mostly very close to

1, with the exception of fine aerosol mode, where they show slightly lower values. Besides, MicroCarb exhibits varying information content on optical path length related variables, depending on whether the 0.76 μm and/or the 1.27 μm $O_2$ sensitive bands are included in the calculations. Regarding NanoCarb, that only measures truncated interferograms and not full spectra, its pixel-wise $X_{CO_2}$ performance compares on many aspects to the performance of low spectral resolution and low-SNR spectra-observing concepts. However, the $X_{CO_2}$ precision obtained after combining several observations of the same location is close to CO2M $X_{CO_2}$ precision. The comparison between NanoCarb DOFs and those of spectra-measuring concepts (regardless of its characteristics) highlights that further improvements of the concept are needed, to increase its information content for interfering geophysical variables.

Given its scope focused on exploring the impact of concept design parameters on $X_{CO_2}$ retrieval performance, this study could not include all the dimensions of a comprehensive mission performance assessment. For example, the accuracy of $X_{CO_2}$ retrieval has not been studied, and a greater variability of possible atmospheric conditions (different aerosol types, layers, contents, etc., different thermodynamical profiles and $CO_2$ concentration vertical profiles) could be encompassed, as is usually performed in comprehensive Observing System Simulation Experiments. Besides, this work could not also obviously explore the whole extent of possible design parameters (e.g. band-wise variations of spectral sampling ratios, varying wavelength interval for spectral bands, combination of different instruments, etc.) that impact $X_{CO_2}$ retrieval performance, and its implication for anthropogenic plume imaging. These limitations warrant further studies.

This new opening decade will see a large increase in spaceborne monitoring of $X_{CO_2}$ from a wide variety of SWIR-observing concepts. This work enabled to explore three of the most critical parameters, and it already shows how different we can expect upcoming $X_{CO_2}$ products will be, in their respective performance and sensitivities to interfering variables. This hints at the extent of work that will be required to compare, reconcile and cross-calibrate the results produced by so many different satellites concepts, especially if their purpose is to support independent evaluation of mitigation efforts aiming at Paris Agreement objectives.

**Data availability**

The outputs of radiative transfer simulations and scripts to generate synthetic performance results are available from Matthieu Dogniaux upon request by email (M.Dogniaux@sron.nl).

**Author contributions**

MD designed this study, and carried it out with the help and supervision of CC. MD wrote this article with feedback from CC.

## Competing interests

The authors declare that they have no conflict of interest.

## Financial support

MD was funded by Airbus Defence and Space in the framework of a scientific collaboration with École polytechnique. This work has received funding from CNES and CNRS. The Space CARBon Observatory (SCARBO) project, that supported NanoCarb development, received funding from the European Union's H2020 research and innovation program under grant agreement No 769032.

## Acknowledgements

The authors are thankful for the funding received from Airbus Defence and Space, CNES and CNRS, and grateful to Raymond Armante and Vincent Cassé for their helpful comments and proof-reading of this manuscript.

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
