# Peer review of "Synthetic mapping of XCO2 retrieval performance from shortwave infrared measurements: impact of spectral resolution, signal-to-noise ratio and spectral band selection"

_Atmospheric Measurement Techniques, 2023_

## Author Comment (AC1)

We are grateful to the referee for this detailed feedback and comments. Our answers are included within the referee's text in red.

Review of "Synthetic mapping of XCO2 retrieval performance from shortwave infrared measurements: impact of spectral resolution, signal-to-noise ratio and spectral band selection" by Dogniaux and Crevoisier, amt-2023-233

The present manuscript presents a simulation study of the impact of spectral resolution, spectral band selection and signal-to-noise (SNR) ratio on XCO2 retrieval performance for space borne passive spectrometers in the shortwave infrared, with a focus on the CO2M, MicroCarb and NanoCarb sensors. A large part of aforementioned parameter space is explored synthetically by simulating top-of-atmosphere radiances with the 5AI radiative transfer model and then operating an optimal estimation based inversion on the synthetic measurements to discuss XCO2 retrieval performance with respect to random errors, vertical sensitivity and information content among others. The paper addresses relevant questions within the scope of AMT and presents new data that warrant publication. The language is fluent, the authors give credit to related work and substantial conclusions are reached. However, I believe that the manuscript could be improved by addressing some methodological questions which should strengthen the interpretation of the results.

General comments:

G1: Recent studies have shown that coarse-resolution SWIR instruments tend to be affected by strong trace gas biases induced by surface albedo features (e.g. for XCO2 or XCH4). I feel that this is not sufficiently addressed in your discussion of retrieval performance as a function of spectral resolution and SNR (section 4.1), in the discussion of information entanglement (section 4.2.3) and your observations regarding XCO2 precision as a function of Degrees of Freedom for Signal (DoFS): In your synthetic study it may be true that precision can be easily "bought" at the expense of spectral resolution by increasing SNR. However, in the real world, for resolving powers below ~1000 I would expect that this stops being true. It would probably go beyond the scope of this paper to include a more realistic albedo model in your framework, but it sure would be interesting. You should at least discuss this more in-depth throughout the manuscript (e.g. near Figure 3, etc.).

We agree with the referee, only albedo models that are assumed to be perfectly known are considered here, thus underestimating the impact of albedo spectral-dependence misknowledge on CO2 (and GHG in general) enhancement retrieval for low resolving power instruments. These effects have for example been discussed by Ayasse et al. (2018). This could indeed be incorporated in the study by either studying the retrieval sensitivity to realistic albedo spectral features or by inflating the Se matrix appropriately to reflect forward model misknowledge. However, as the referee noted, this would go beyond the scope of the paper which – at this stage – focuses on SNR-based performance results only.

Besides, this remark could also be generalized to all resolving power, for which we assume, for example, that spectroscopic parameters are perfectly known, as well as aerosol optical properties.

In the revised manuscript (1) we better precise the scope of the study regarding what is included in the performance; (2) we elaborate better on how the Se matrix is constructed and

the implications of these choices; and (3) we include additional discussion elements dedicated to the limits of this study regarding the precision of low-SNR observing concepts.

| Lines 106-108 | In this work, we perform a systematic survey that synthetically explores the impact **of spectral resolution, signal-to-noise ratio (SNR) and spectral band selection, three design parameters for SWIR CO$_2$ observing satellite concepts,** on $X_{CO_2}$ retrieval performance **(SNR-related precision, degrees of freedom and smoothing error,** excluding accuracy**)** |
|---|---|

| Lines 328-346 | with $\sigma_e$, the noise model for a given spectral sample, and $L$, the radiance for a given spectral sample. **Here, the $S_e$ matrix only includes measurement noise, so the uncertainty (or precision) results obtained from the $\widehat{S}$ matrix will only be related to measurement noise. In practice smoothing errors from CO$_2$ and non-CO$_2$ state vector parameters add up to the uncertainty (Connor et al., 2008). Finally, uncertainty evaluated on real data is typically larger than the uncertainty obtained from Optimal Estimation calculations, as model-related errors (misknowledge of albedo spectral dependence, spectroscopy errors, etc.) are also encompassed in such evaluations. For example, at low resolving powers, the complexity of spectral dependence in surface reflectance can lead to significant errors (e.g. for methane, Ayasse et al., 2018) that will not be accounted for here or, at high resolving power, real-data uncertainties are evaluated to be twice as large as theoretical uncertainty for OCO-2 (Eldering et al., 2017).** |
|---|---|

G2: The fact that turning the aerosol fit off produces strong albedo correlations for NanoCarb indicates to me that you might be overfitting and/or need to adjust the retrieval scheme for NanoCarb. Perhaps the aerosol parameters are simply within the null space. If we think of NanoCarb as a low resolving power spectrometer, it seems physically very plausible that aerosol and albedo parameter fits would strongly impact each other. Please double check the retrieval and discuss this observed behavior. Which retrieval setting is recommended for NanoCarb? How do the retrieved values and standard deviations of XCO2 and albedo change when you turn the aerosol fit off and on?

This point raised by the referee about NanoCarb null space is interesting. Following this idea, we performed a Singular Vector Decomposition (SVD) of the NanoCarb Jacobian matrix with its columns scaled for state vector perturbations equal to a priori uncertainty. We have: y = Kx = UAV$^T$x, with the columns of V giving a basis for the row space. Figure R1.1 shows the obtained singular value distribution with noise level for all four NanoCarb bands. There are 28 singular vectors (numbered from 0 to 27) as the state vector for NanoCarb is 28-variables long.

[Figure]

*Figure R1.1. Singular value distribution for NanoCarb Jacobian matrix scaled for perturbations equal to a priori uncertainty, for the VEG 50° observational situation. Colored lines show the noise level for all four NanoCarb bands.*

While we can numerically compute a row space basis with 28 dimensions, the analysis of the singular value distribution against noise levels shows that a maximum of 8 singular vectors are actually significant (singular vector numbers from 0 to 7 in Fig. R1.1). The remaining 20 vectors are related to singular values lower than the noise level and thus constitute a basis for the null space of NanoCarb Jacobian matrix. First, Figure R1.2 shows significant singular vectors of the row space basis.

[Figure]

[Figure]

*Figure R1.2. The seven first significant singular vectors of NanoCarb row space, for the VEG 50° observational situation.*

Interestingly, but unsurprisingly, we find that the first four vector are each overall related to the albedo value of one particular band (with very small non-zero contributions of aerosol parameters). Non-zero contributions of CO2 concentration, water vapor, temperature shift, surface pressure and coarse and fine optical depths are present in the following four vectors. As we can see, coarse mode aerosol shows significant contributions to the 6[th] and 7[th] singular

vectors, and fine mode aerosol small contributions to all singular vectors shown in Fig. R1.2. Interestingly, looking into less-significant singular vectors, we find that the 10th vector (numbered 9 in Fig. R1.1) shows a significant contribution from fine mode aerosols (see Fig. R1.3). Consequently, we conclude that while both coarse and fine mode do not completely lie in NanoCarb null space, fine mode is indeed considerably lying into it. This analysis is consistent with DOFS values obtained for aerosol parameters as show in Fig 6 of the main text.

[Figure]

*Figure R1.3. The 9th singular vector of the row space basis (associated to a singular value lower than noise level) obtained through SVD decomposition of NanoCarb Jacobian matrix.*

Indeed, as seen in the first four singular vectors, we have small non-zero contributions of aerosol parameters in these vectors, thus showing entanglement of albedo and aerosol parameters. This can also be seen in the covariance matrices shown in Fig. 10 of the main text.

NanoCarb is a compact novel observing concept which performance, merits and drawbacks, especially compared to "classic" spectra-observing instruments, are still under examination. This work builds upon the NanoCarb performance assessment performed in Dogniaux et al, 2022. The definition of the appropriate state vector for NanoCarb is thus still in consideration. As one of the goals of this study is to compare NanoCarb performance against "classic" spectra-observing concepts, we decided to include aerosol parameters in the state vector for NanoCarb in order to ensure a comparison as-fair-as-possible. Indeed, we agree with the referee that it would certainly make sense not to include these geophysical parameters in the state vector considering how little information appears to be, as it would make sense for low resolving power cases in the CVAR experiment (see question G3 below).

As the main goal of this work is to assess the impact of instrument design and characteristics on XCO2 retrieval performance, we chose to use an as-similar-as-possible state vector throughout all cases of CVAR and for exactly-defined concepts (including NanoCarb) in order to ensure an as-fair-as-possible comparison. We propose to better precise the goal of our state vector composition with respect to the goals of this study in the revised manuscript.

| Lines 321-325 | We include in the state vector all the main geophysical variables necessary to model SWIR spaceborne measurements. Those are listed in Table 4, along with their a priori values and uncertainties. **This selection of state vector parameters is used for all exactly-defined concepts (with a small exception on albedo slope for NanoCarb) and CVAR experiment cases because the goal of this study is to explore the impact of design parameters on performance. Using a similar estimation scheme across all resolving power helps us achieve this goal. However, we do realise that, in practice, less geophysical elements are fitted for low resolving power observations (Cusworth et al., 2021).** |
|---|---|

Besides, as suggested by the referee, messages on NanoCarb could also be expanded regarding recommending a relevant state vector composition for NanoCarb.

| Lines 730-732 | … as well as non-zero yet rather low DOF values for $H_2O$ profile scaling factor, temperature global shift and coarse mode aerosol optical depth (0.93, 0.36 and 0.55, respectively). **Given how little some of these DOF values are, we also conclude that the state vector used to process NanoCarb measurement should be adjusted to only include the most essential geophysical variables. For example, aerosol optical depths could be removed.** |
|---|---|

As we only compute $\hat{S}$ for our performance calculation (explained at the beginning of Sect. 4.1.1), there is no retrieved value available for XCO2 and albedo in our scheme. However, we can indeed examine how their standard deviations change when the aerosol fit is turned on and off. These results are given for the VEG 50° observational situation in Table R1.1.

| | Std deviations **with** aerosol fitting | Std deviations **without** aerosol fitting |
|---|---|---|
| XCO2 | **5.572 ppm** (0.600 ppm) | **5.566 ppm** (0.600 ppm) |
| Albedo B1 | **6.83x10$^{-2}$** (6.80x10$^{-3}$) | **4.87x10$^{-3}$** (5.33x10$^{-4}$) |
| Albedo B2 | **2.20x10$^{-2}$** (2.18x10$^{-3}$) | **3.94x10$^{-4}$** (3.87x10$^{-5}$) |
| Albedo B3 | **2.31x10$^{-2}$** (2.28x10$^{-3}$) | **3.52x10$^{-4}$** (3.46x10$^{-5}$) |
| Albedo B4 | **5.42x10$^{-3}$** (5.33x10$^{-4}$) | **9.32x10$^{-4}$** (9.42x10$^{-5}$) |

*Table R1.1. A posteriori standard deviations obtained with and without fitting aerosols, for single-pixel retrieval (in bold) and combined results for different viewing angles (in parenthesis), for the VEG 50° observational situation.*

As could be expected, we find lower posterior standard deviations for albedo parameters when turning off the fitting of aerosol parameters. XCO2 posterior standard deviation decreases as well, but very little. In conclusion, aerosol parameters and albedo parameters do

impact each other. We can also see that while fitting the optical depth of two aerosol layers may indeed be a stretch for NanoCarb, given the current state vector design, this choice has nonetheless very little impact on reported XCO2 standard deviation values.

G3: How does the above observation extend to the CVAR experiment? Do you fit aerosol parameters across the spectral-resolution space that you cover?

We do fit aerosol across the spectral-resolution space that is covered in the study. Figure R1.4 shows how standard deviations of XCO2 and albedo related parameters (for the observational situation VEG 50°) evolve when turning on and off aerosol fitting.

[Figure]

*Figure R1.4. XCO2 (top left) and albedo baseline (top center and right) and band-wise slopes (bottom center and right) standard deviations obtained with (full lines) and without (dashed lines) fitting aerosol parameters. For better readability, the bottom left panel shows the ratio between without and with the fitting of aerosol parameters for XCO2 precision.*

Similarly to the results obtained with NanoCarb, XCO2 standard deviations reduce by a very little amount (less than 10%) compared to the standard deviations of albedo-related parameters (about an order of magnitude for baseline albedo and from 10% to an order of magnitude for band-wise albedo slope). Thus, results currently presented for CVAR while fitting aerosol parameters are quite representative of what we would obtain without fitting aerosol parameters.

If yes, would it not make sense to turn off the fine mode parameter at some low resolving power (see Fig. 6)?

If we had to operationally process low-resolving power observations ($\lambda/\Delta\lambda < 1000$), it would indeed make sense to turn off fitting fine mode aerosols, and maybe even coarse mode. Usually, these observations can even be processed using a matched filter technique that basically just fits the target atmospheric species (e.g. Guanter et al., 2021). However, as explained before for NanoCarb, the goal is to exhibit the roles of spectral resolution and SNR in XCO2 retrieval performance, thus we seek an as-fair-as-possible comparison across all cases of CVAR and exactly-defined concept.

As detailed for question G2 regarding NanoCarb, we adjusted in the following manner the state vector composition presentation:

| Lines 321-325 | We include in the state vector all the main geophysical variables necessary to model SWIR spaceborne measurements. Those are listed in Table 4, along with their a priori values and uncertainties. **This selection of state vector parameters is used for all exactly-defined concepts (with a small exception on albedo slope for NanoCarb) and CVAR experiment cases because the goal of this study is to explore the impact of design parameters on performance. Using a similar estimation scheme across all resolving power helps us achieve this goal. However, we do realise that, in practice, less geophysical elements are fitted for low resolving power observations (Cusworth et al., 2021).** |
|---|---|

I am surprised that you appear to find sufficient DoFS for the coarse mode parameter even at a resolving power of 200 - how do you explain the very different behavior of the two aerosol retrieval parameters?

In this retrieval scheme, we assume that aerosol optical properties are perfectly known, including the spectral dependence of these optical properties. Figures R1.5 and R.16 show relative radiance sensitivities to aerosol optical depth change for a resolving power of 200 and 10000, respectively.

[Figure]

*Figure R1.5. Radiance (black, left y-axis) and relative sensitivities to a 0.05 optical depth increase for coarse (red, right y-axis) and fine (blue, right y-axis) modes for all CVAR bands B1, B2 and B3 (top, center and bottom panels, respectively), for the VEG 50° at λ/Δλ=200.*

[Figure]

*Figure R1.6. Radiance (black, left y-axis) and relative sensitivities to a 0.05 optical depth increase for coarse (red, right y-axis) and fine (blue, right y-axis) modes for all CVAR bands B1, B2 and B3 (top, center and bottom panels, respectively), for the VEG 50° at λ/Δλ=10000.*

For both resolving power cases, we can notice that coarse mode aerosols produce relative sensitivities with stronger spectral features than fine mode aerosols. The DOFS obtained for coarse mode aerosols at very low resolving power may be explained by these stronger spectral features that bring specific information to the estimation scheme, unlike for fine mode aerosols.

G4: As far as I can tell, the paper currently does not address the accuracy of XCO2 retrievals as a function of SNR, resolving power and band selection. I believe it would be worthwhile to include this.

Indeed, the scope of this work only includes performance as a posteriori uncertainty given by the $\hat{S}$ matrix (Sect. 4.1.1), and explores smoothing error caused by prior misknowledge of a priori interfering parameters (Sect. 4.2.3 and 4.2.4). Including an actual accuracy assessment would indeed be worthwhile, but would require to extend the scope too much to fit within this publication only, as it would tend towards an actual OSSE. This question could be however be explored in a follow-up study. To manage reader expectations, we propose to better precise the goals of this study and acknowledge this limit in the conclusions, suggesting such a follow-up study on accuracy evaluation.

| Lines 106-108 | In this work, we perform a systematic survey that synthetically explores the impact **of spectral resolution, signal-to-noise ratio (SNR) and spectral band selection, three design parameters for SWIR CO₂ observing satellite concepts,** on $X_{CO_2}$ retrieval performance **(SNR-related precision, degrees of freedom, vertical sensitivity and smoothing error, excluding accuracy)**. |
| --- | --- |

| Lines 842-849 | **Given its scope focused on exploring the impact of concept design parameters on $X_{CO_2}$ retrieval performance, this study could not include all the dimensions of a comprehensive mission performance assessment. For example, the accuracy of $X_{CO_2}$ retrieval has not been studied, and a greater variability of possible atmospheric conditions (different aerosol types, layers, contents, etc., different thermodynamical profiles and CO₂ concentration vertical profiles) could be encompassed, as is usually performed in comprehensive Observing System Simulation Experiment. Besides, this work could not also obviously explore the whole extent of possible design parameters (e.g. band-wise variations of spectral sampling ratios, varying wavelength interval for spectral bands, combination of different instruments, etc.) that impact $X_{CO_2}$ retrieval performance, and its implication for anthropogenic plume imaging. These limitations warrant further studies.** |
| --- | --- |

Minor comments:

M1: In Figure 1, it would be helpful to indicate the FWHM of the band-pass filters of NanoCarb and remind the reader of the FWHM of the NanoCarb bands.

We included these pieces of information in Figure 1 and the FWHM values in the text.

| Line 232 | … It measures truncated interferograms that are sensitive to four spectral bands, as shown in Fig. 1 right panel, which are associated to narrow-band filters shown in Fig. 1 left panel (follow the arrows). **Their FWHMs are 35 cm⁻¹, 24 cm⁻¹, 69 cm⁻¹ and 18 cm⁻¹ for band 1 to 4, respectively.** NanoCarb has a two-dimensional field-of-view… |
| --- | --- |

M2: Section 3.1: Please explain in the text why you confine aerosol particles to the lowermost 4 km of the atmosphere and add details on what kind of aerosol size distributions you used.

This choice is based on Table 2 included in Papayannis et al. (2008) showing that transported desert dust over Europe has a center of mass typically around 3000-3800 m in altitude with layer base altitude values overall comprised between 2000-3000 m and layer top altitude values overall comprised between 4000-5000 m. Papayannis et al. (2008) also show how

variable the layer of transported dust can be and it would definitely be interesting to explore various aerosol content conditions in follow-up studies.

In the revised manuscript, we add the Papayannis et al. (2008) reference and expand on possible further work in the conclusions.

| Lines 265-266 | … coarse mode aerosols, representative of minerals, are included between 2 and 4 km of altitude **(this choice is supported by transported desert dust layers over Europe described by Papayannis et al., 2008)**. |
|---|---|

| Lines 843-846 | **For example, the accuracy of $X_{CO_2}$ retrieval has not been studied, and a greater variability of possible atmospheric conditions (different aerosol types, layers, contents, etc., different thermodynamical profiles and $CO_2$ concentration vertical profiles) could be encompassed, as is usually performed in comprehensive Observing System Simulation Experiment.** |
|---|---|

Regarding aerosol size distribution, the OPAC library software uses lognormal size distributions. We added this information in the main text.

| Lines 313-314 | … aerosol optical properties are taken from the OPAC library**, which uses lognormal size distributions** (Hess et al., 1998). |
|---|---|

M3: Section 3.1: The albedo values you chose for the SWIR seem pretty high in my opinion. Is this really the value over all ASTER samples?

These three albedo models used here were the same that have been used for the initial NanoCarb L2 performance assessment in Dogniaux et al. (2022). They were built from the ASTER spectral library (by co-authors who contributed to Dogniaux et al., 2022), with the 'soil' (SOL) model intended as a minimum albedo case and the 'desert' (DES) model intended as a maximum case. Figure R1.7 taken from Dogniaux et al. (2021) shows all vegetation ASTER albedo models (thin grey lines) along SOL, VEG and DES albedo models used here, and with retrieved albedo values for all three OCO-2 bands at different TCCON stations (markers). We can see that the retrieved albedo values for OCO-2 bands satisfyingly match values corresponding to a mixture of SOIL and VEG albedo models, as expected considering where TCCON stations are located on Earth. Thus, we think that these three albedo models are realistic enough for the scope of the performance assessment

[Figure]

*Figure R1.7. Retrieved albedo values for all three OCO-2 bands (markers, distributed over ±0.1 μm intervals around actual OCO-2 band locations for readability) for different TCCON stations and OCO-2 target sessions and all ASTER vegetation models (thin grey lines). The models used in this study are shown in colored lines. Figure taken from Dogniaux et al (2021).*

M4: Figure S1: It would be useful to magnify the albedo Jacobian for NanoCarb, especially in the context of albedo biases and aerosol retrievals (see above).

We have split Fig. S1 between CO2M and NanoCarb (Fig. S2 in revised Supplements) to provide figures that are easier to read.

M5: Page 7, line 147: add reference for CO2M XCO2 random error value of 0.7 ppm

We have added a reference to Meijer 2020, the CO2M Mission Requirement Document: https://esamultimedia.esa.int/docs/EarthObservation/CO2M_MRD_v3.0_20201001_Issued.pdf

| Line 179 | … aiming to provide an imaging of $X_{CO_2}$ with a random error lower than 0.7 ppm and systematic errors as low as possible **(Meijer, 2020).** |
|---|---|

M6: Page 9, line 189: add reference for the "latest design of the NanoCarb instrument"

We have added the reference

| Lines 227-230 | … we use here the latest design of the NanoCarb instrument, **currently considered with a ~200 km swath and a 2.3 x 2.3 km2 spatial resolution. The optimal OPD selection accounts for CO2 information entanglements with H2O and aerosols, neglects** |
|---|---|

| | **atmospheric temperature and assumes that albedo is band-wise constant (Gousset et al., 2019).** |
|---|---|

M7: Section 2.3: Mention a few more details on NanoCarb so the reader doesn't have to go to other references to find out about the ground sampling distance, swath, etc.

We have added information on swath and spatial resolution in the NanoCarb presentation.

| Line 227 | As in Dogniaux et al. (2022), we use here the latest design of the NanoCarb instrument, **currently considered with a ~200 km swath and a 2.3 x 2.3 km² spatial resolution.** |
|---|---|

M8: Section 3.2.2: Why did you use the GEISA 2015 database as opposed to HITRAN (and as opposed to a newer version)?

We chose the GEISA spectroscopic database that has been selected by the French Centre National d'Études Spatiales (CNES) as the reference spectroscopic database for different satellite instruments, including the (thermal) Infrared Atmospheric Sounding Interferometer (IASI) and MicroCarb. Regarding shortwave infrared, Dogniaux et al. (2021) showed that it is possible to obtain fair L2 retrieval results (against TCCON and ACOS columns) from OCO-2 spectra.

Here, GEISA 2015 is used instead of GEISA 2020 because the set of radiative transfer simulations used in this work has been performed while rolling out the newer GEISA 2020 version (Delahaye et al., 2021). We do not expect significant changes for synthetic performance results between both versions.

M9: Section 3.2.2: How about the treatment of the 1.27 um band and airglow emission? Did you leave it out or did you model it for MicroCarb?

We do not account for Airglow emission in the 1.27 μm band. We added this information in the revised manuscript.

| Lines 317-318 | Finally, the atmospheric model is discretized in 20 atmospheric layers that bound 19 layers, as done by the ACOS algorithm (O'Dell et al., 2018). **Airglow emission which impacts MicroCarb 1.27 μm band (Bertaux et al., 2020) is not included here in 4A/OP simulations.** |
|---|---|

M10: Table 4: To which parameter does the following statement refer? "Not included in the state vector for NanoCarb". Please clarify.

This statement refers to 1st order spectral dependence parameter of band-wise albedo models (the band-wise spectral slope of the albedo). It has been left out from the state vector for NanoCarb because its latest design (Gousset et al, 2019) assumed that albedo is band-wise constant, and NanoCarb measurements do not carry information to satisfyingly fit the albedo slope, despite its impacts on retrieved XCO2 values (not shown). Upcoming developments will explore adjustments to the NanoCarb OPD selection and design to enable the fitting of the spectral dependence of albedo over the wavelengths observed by NanoCarb bands.

In the revised manuscript, Table 4 now includes a reference to Sect 2.3 in this cell, where we extended the text to better explain that the OPD NanoCarb selection was performed assuming constant band-wise albedos.

| Lines 328-330 | **The optimal OPD selection** accounts for $CO_2$ information entanglements with $H_2O$ and aerosols, **neglects a**tmospheric temperature **and assumes that albedo is band-wise constant** (Gousset et al., 2019). |
|---|---|

| Line 347 (Table 4) | Not included in the state vector for NanoCarb case**, see Sect. 2.3.** |
|---|---|

M11: Page 14, lines 311-313: The XCH4 biases in Borchardt et al. (2021) are caused by albedo interferences, not by the prior information. Please reword.

Actually, here, we do not refer to the albedo-related interferences reported in Borchardt et al. 2021, but to the – included – vertical averaging kernel correction that is mentioned page 1270 (right column) of their article's pdf. Citing here:

"The light passed the air mass above the air- craft once on the downward path to the Earth but transected the air mass below the aircraft on both downward and up- ward paths. Consequently, the retrieval was more sensitive to atmospheric changes below the aircraft than above. This was captured by the averaging kernel which represented the sensitivity of the instrument to changes at a specific altitude layer. In our case, strong local enhancements in atmospheric methane were confined below the aircraft, so we multiplied the total column enhancements by the inverse of the averaging kernel for the air mass underneath the aircraft (kAK) to determine the true enhancement of CH4 caused by an emission source near the ground."

This correction is described in Krings et al, 2011, in equations 11 and 12. The point we make here is that, similarly to aircraft-based retrievals, the vertical-sensitivity of satellite-based XCO2 retrievals must be examined and any non-unity sensitivity must be corrected for.

We revised the sentence to better underline that we refer to artefacts caused by non-unity vertical sensitivity, and how the given references relate to this question.

| Line 432 | Indeed, any deviation from **unity** in the vertical sensitivity wrongfully scales differences between the unknown truth and the prior into the retrieved column enhancement, thus calling for a posteriori corrections, **as presented by (Krings et al., 2011) and also included by (Borchardt et al., 2021) for aircraft observation processing.** |
|---|---|

M12: Figure 8: This is related to G1. Your albedo perturbation is not doing anything to the retrieved XCO2 under any scenario. I think you should introduce a linear or periodic perturbation and observe the effect. This would be a very interesting experiment!

The albedo perturbation is not effect-less, see Figure R1.8 that provides a Zoom for the albedo part of Figure 8, but these effects are very small indeed.

[Figure]

*Figure R1.8. Zoomed in the albedo-related section of Figure 8.*

We agree that including a linear or periodic perturbation to the albedo would be an interesting experiment. This could be explored in a follow-up study exploring a wide variety of observational situations tending towards the OSSE. Besides, as we cannot fairly explore the effect of albedo spectral dependence with the NanoCarb concept given its design assumptions (see answer to M10), we refrain from addressing this question for the moment.

M13: Figure 3: Maybe add a "disclaimer" to the caption, explaining that NanoCarb is symbolically added at very low resolving power (as in Page 23, lines 503-504).

We added this disclaimer in the revised manuscript for all figures that include NanoCarb markers against spectral resolution. Thank you for this suggestion. (Figures 3, 5 and 6, as well as Figures S4, S5, S10-17)

M14: Page 27, lines 607-608: I am not convinced that this statement is generally true across the resolving power space you explore. Potentially reword.

For reference, the statement lines 607-608 is: "Overall, precision is more easily gained through SNR improvement than through spectral resolution improvements."

This statement is indeed ambiguous as it does not precise which scale of change is considered here. In the original manuscript, we meant precision gains for the maximum range of change available in the 2-dimensional space (SNR x resolving power) that we explore. Figure R1.9

shows in full lines the Normalized XCO2 precision improvement for a resolving power (resp. SNR) improvement of 2.2 orders of magnitude (resp. 2.) for all SNR (resp. resolving power) values in red (resp. blue). We note this Normalized XCO2 precision improvement with $G$, and have:

$$G = \left(\frac{\sigma_{XCO2}(x_i)}{\sigma_{XCO2}(x_j)}\right) / (\frac{x_j}{x_i})$$

With $\sigma_{XCO2}$ the XCO2 precision and $x$, the variable along which we study the improvement, with $x_i < x_j$.

[Figure]

*Figure R1.9. Normalized XCO2 precision improvement (G) for a resolving power (resp. SNR) improvement of different magnitudes (line styles) for all SNR (resp. resolving power) values in red (resp. blue).*

We added some precisions in the revised manuscript.

| Line 793-794 | Overall, for these large changes **of about two orders of magnitude**, precision **is more sensitive** to SNR improvement than to spectral resolution improvements. |
|---|---|

However, as the referee guessed, this statement is not true for all ranges of change. Dashed and dotted lines in Figure R1.9 show Normalized XCO2 precision improvements for changes of an order of magnitude (or slightly more). It is interesting to note that an order of magnitude of improvement in resolving power shows similar improvement in XCO2 precision as an order of magnitude of improvement in SNR for the higher end of explored resolving power values (dotted red line) only (to be compared to the dashed red line). This is due to the $\lambda/\Delta\lambda$ $\sim 1000 - 4000$ resolving power range where increasing spectral resolution is less efficient to improve XCO2 precision compared to elsewhere in the explored, as it can be noted on Figure R1.10, that shows local Normalized XCO2 precision improvement values.

[Figure]

*Figure R.10. Local Normalized XCO2 precision improvement along resolving power (top left) and SNR (top right), and the ratio of these improvements (along resolving power divided by along SNR)*

We added some text in the revised manuscript to note this interesting role of the $\lambda/\Delta\lambda \sim 1000 - 4000$ range and added Fig. R1.10 to the Supplements.

| Line 392-396 | Hence, it appears that $X_{CO_2}$ precision **is more sensitive to** SNR improvements **rather than to** resolving power improvements, for large improvements of two orders of magnitude centred on $CO_2M$ instrument characteristics. **However, as it can be seen in Fig. 3 (and in Supplementary Fig. S3), this conclusion does not hold for smaller local improvements, that generally result in better $X_{CO_2}$ precision gains through resolving power improvements than through SNR improvements (see Supplementary Fig. S3).** |
|---|---|

| Lines 402-407 | it corresponds to the individual spectral lines of $CO_2$ becoming **clearly** visible in spectral band branches, as also commented for Fig. 2. **Between these two slope breaks ($\lambda/\Delta\lambda \sim 1000 - 4000$), improvements in resolving power are less efficient in improving $X_{CO_2}$ precision than elsewhere along the resolving power dimension (see Supplementary Fig. S3). This explains why, across the large two order of magnitude improvements in resolving** |
|---|---|

| | power and SNR explored in this study, SNR has a larger impact on precision than resolving power. This result also underlines the critical importance of resolving new spectral features (the P-R band structure below $\lambda/\Delta\lambda \sim 1000$ or the individual spectral lines above $\lambda/\Delta\lambda \sim 4000$) to gain $X_{CO_2}$ precision efficiently. |
|---|---|

We consequently brought nuance into the conclusions:

| Lines 793-795 | Overall, **for these large changes of about two orders of magnitude**, precision is more easily gained through SNR improvement than through spectral resolution improvements. **However, we see that small magnitude improvements in resolving power generally yield more $X_{CO_2}$ precision improvements than SNR improvements, especially when CO₂ spectral lines are resolved.** The separate |
|---|---|

M15: I find that you could try to summarize your results more concisely in the conclusion: e.g. The SNR-resolving power-space holds very little performance gain for sensors with two SWIR bands, the 2 um CO2 band carries higher XCO2 and aerosol DoFS than the 1.6 um band, etc…

We have removed some less essential details from the conclusions (see tracked-changes version of the MS Word document).

Technical comments:

T1: check entire reference list for spelling errors and typesetting issues

We have corrected unresolved LaTeX-related typos not handled by the MS Word Zotero plugin.

T2: Abstract, line 22: perturbating -> perturbing

Corrected, thank you for catching this spelling mistake.

T3: Page 2, line 35: …carbon cycle compares results… -> …carbon cycle is based on comparisons of results…

Corrected, thank you.

 T4: Page 2, line 44: pioneer -> pioneering

Corrected, thank you.

T5: Page 2, line 59: gather -> are responsible for

Corrected, thank you.

T6: Page 3, line 64: atmospheric schemes -> atmospheric inversion schemes ?

Corrected, thank you.

T7:  Page 7, line 165: measure a continuous spectra -> measure continuous spectra

 Corrected, thank you.

T8: Page 11, line 246-247: A also enables to compute the … -> A also enables computation of …

Corrected, thank you.

T9: Page 11, line 255: layers -> levels

As "levels" is used is the original manuscript and not "layers", we assume that the referee suggested a change in the other way around, and indeed "layers" is more appropriate than "levels".

T10: Page 15, line 336: "which sensitivities also correlate" reword

| Line 467 | This spectral band is indeed sensitive to surface pressure, temperature and aerosols, and can thus bring independent constraint on these geophysical parameters, **that show** sensitivities **that** correlate with $CO_2$ sensitivity in 1.6 and 2.05 µm bands |
|---|---|

 T11: Page 23, line 516: variable Jacobians -> Jacobian variables ?

We refer to both CO2 and other variables' partial derivative, so the sentence seems correct as it is. We replace Jacobians to may be ease the understanding:

| Line 685 | One can indeed notice that $CO_2$ and other variable**s' partial derivatives** are correlated |
|---|---|

**References**

Alana K. Ayasse, Andrew K. Thorpe, Dar A. Roberts, Christopher C. Funk, Philip E. Dennison, Christian Frankenberg, Andrea Steffke, Andrew D. Aubrey, Evaluating the effects of surface properties on methane retrievals using a synthetic airborne visible/infrared imaging spectrometer next generation (AVIRIS-NG) image, Remote Sensing of Environment, Volume 215, 2018, Pages 386-397, ISSN 0034-4257, https://doi.org/10.1016/j.rse.2018.06.018.

Dogniaux, M., Crevoisier, C., Gousset, S., Le Coarer, É., Ferrec, Y., Croizé, L., Wu, L., Hasekamp, O., Sic, B., and Brooker, L.: The Space Carbon Observatory (SCARBO) concept: assessment of $X_{CO2}$ and $X_{CH4}$ retrieval performance, Atmos. Meas. Tech., 15, 4835–4858, https://doi.org/10.5194/amt-15-4835-2022, 2022.

Luis Guanter, Itziar Irakulis-Loitxate, Javier Gorroño, Elena Sánchez-García, Daniel H. Cusworth, Daniel J. Varon, Sergio Cogliati, Roberto Colombo, Mapping methane point emissions with the PRISMA spaceborne imaging spectrometer, Remote Sensing of

Environment, Volume 265, 2021, 112671, ISSN 0034-4257, https://doi.org/10.1016/j.rse.2021.112671.

Papayannis, A., et al. (2008), Systematic lidar observations of Saharan dust over Europe in the frame of EARLINET (2000–2002), J. Geophys. Res., 113, D10204, doi:10.1029/2007JD009028.

Matthieu Dogniaux. Suivi de la concentration atmosphérique de CO2 par satellite : performances et sensibilités des prochains concepts d'observation dans le proche infrarouge. Ingénierie de l'environnement. Institut Polytechnique de Paris, 2021. Français. ⟨NNT : 2021IPPAX119⟩. ⟨tel-03662470⟩, https://theses.hal.science/tel-03662470

Dogniaux, M., Crevoisier, C., Armante, R., Capelle, V., Delahaye, T., Cassé, V., De Mazière, M., Deutscher, N. M., Feist, D. G., Garcia, O. E., Griffith, D. W. T., Hase, F., Iraci, L. T., Kivi, R., Morino, I., Notholt, J., Pollard, D. F., Roehl, C. M., Shiomi, K., Strong, K., Té, Y., Velazco, V. A., and Warneke, T.: The Adaptable 4A Inversion (5AI): description and first $X_{CO_2}$ retrievals from Orbiting Carbon Observatory-2 (OCO-2) observations, Atmos. Meas. Tech., 14, 4689–4706, https://doi.org/10.5194/amt-14-4689-2021, 2021.

T. Delahaye, R. Armante, N.A. Scott, N. Jacquinet-Husson, A. Chédin, L. Crépeau, C. Crevoisier, V. Douet, A. Perrin, A. Barbe, V. Boudon, A. Campargue, L.H. Coudert, V. Ebert, J.-M. Flaud, R.R. Gamache, D. Jacquemart, A. Jolly, F. Kwabia Tchana, A. Kyuberis, G. Li, O.M. Lyulin, L. Manceron, S. Mikhailenko, N. Moazzen-Ahmadi, H.S.P. Müller, O.V. Naumenko, A. Nikitin, V.I Perevalov, C. Richard, E. Starikova, S.A. Tashkun, Vl.G. Tyuterev, J. Vander Auwera, B. Vispoel, A. Yachmenev, S. Yurchenko, The 2020 edition of the GEISA spectroscopic database, Journal of Molecular Spectroscopy, Volume 380, 2021, 111510, ISSN 0022-2852, https://doi.org/10.1016/j.jms.2021.111510.

Gousset, S., Croizé, L., Le Coarer, E., Ferrec, Y., Rodrigo-Rodrigo, J., Brooker, L., and consortium, for the S.: NanoCarb hyperspectral sensor: on performance optimization and analysis for greenhouse gas monitoring from a constellation of small satellites, CEAS Space J., 11, 507–524, https://doi.org/10.1007/s12567-019-00273-9, 2019

Krings, T., Gerilowski, K., Buchwitz, M., Reuter, M., Tretner, A., Erzinger, J., Heinze, D., Pflüger, U., Burrows, J. P., and Bovensmann, H.: MAMAP – a new spectrometer system for column-averaged methane and carbon dioxide observations from aircraft: retrieval algorithm and first inversions for point source emission rates, Atmospheric Meas. Tech., 4, 1735–1758, https://doi.org/10.5194/amt-4-1735-2011, 2011.

Borchardt, J., Gerilowski, K., Krautwurst, S., Bovensmann, H., Thorpe, A. K., Thompson, D. R., Frankenberg, C., Miller, C. E., Duren, R. M., and Burrows, J. P.: Detection and quantification of $CH_4$ plumes using the WFM-DOAS retrieval on AVIRIS-NG hyperspectral data, Atmospheric Meas. Tech., 14, 1267–1291, https://doi.org/10.5194/amt-14-1267-2021, 2021.

---

## Author Comment (AC2)

We are grateful to the referee for these positive comments and suggestions. We are replying to the comments and questions in red, between the referee's text.

The paper "Synthetic mapping of XCO2 retrieval performance from shortwave infrared measurements: impact of spectral resolution, signal-to-noise ratio and spectral band selection" provides detailed analysis of the characteristics of CO2 measurements with SWIR spectrometers with a range of resolutions, signal to noise characteristics, and spectral bands. Characteristics of actual spectrometer designs (CO2M, MicroCarb, NanoSat), as well as a large set of hypothetical instrument configurations are considered.

The paper reports on detailed analysis of the XCO2 precision, the degrees of freedom for CO2, the vertical sensitivity, the sensitivity for and possible interference due to parameters such as temperature, water vapor, albedo, and aerosols. A wide range of scenarios (or situations) are explored, where surface reflectance and solar geometry are systematically changed.

Overall, this paper is very well constructed. The experiments and assessments are carefully structured and the key findings are clearly described. The graphics are effective, the completeness of the analysis is impressive, and the writing is clear.

General review comments:

Overall, this is an impressive and comprehensive piece of work. It can serve as a reference for instrument developers as they seek to optimize performance and evaluate the trade space of resolution, signal to noise, and band pass. The methodology is clearly described, including the input data and calculations that are performed.

One weakness I find in this paper is the treatment and description of the CO2M instrument. There should be language included to clarify that this work is assessing just the spectrometer element of CO2M, which will also integrate a multi-angle polarimeter. The assessment of XCO2 precision and error related to aerosols is a correct analysis for the CO2M spectrometer alone, but not the planned CO2M mission. I would suggest that this point is made clear at the beginning, and perhaps they use the phrase CO2M spectrometer in the paper.

We agree with the referee that this is indeed a very important point. It was already discussed in the original manuscript when exploring aerosol sensitivities for CO2M, near the article end: "These values are well below the 0.5 ppm systematic error requirement (Meijer, 2020), and are expected to be even more reduced by using the aerosol observations provided by the Multi-Angle Polarimeter that will fly along CO2M spectrometers (e.g. Rusli et al., 2021)."

As MAP results can also help to have reduced prior uncertainty on aerosol parameters, and thus also impact precision results, we agree with the referee that this should point should have also been discussed earlier in the paper. We added some extra comments when presenting CO2M to discuss this point right from the start.

| | |
|---|---|
| Lines 180-184 | In this work, we use the **spectrometer** measurement characteristics presented in Table 3 to model the $CO_2M$ concept. **Besides the spectrometer, the CO₂M mission will also include a Multi-Angle Polarimeter, which is an instrument dedicated to the observation of aerosols. Its results are expected to help better constrain their** |

| | interfering effect on $X_{CO_2}$ retrievals, and improve their precision and accuracy (Rusli et al., 2021). Here, we only study the CO₂M spectrometer alone, thus the results that we obtain do not reflect the comprehensive theoretical CO₂M mission performance. |
|---|---|

To gain confidence in the methodology, it would be useful for the authors to point out where OCO-2 and GOSAT(-2) are in these SNR/resolution plots, and to compare to published results for precision, DOF, etc. I suggest this because sources of error such as spectroscopic mischaracterization or errors in instrument characterization are not well captured in the analysis presented here, yet may be important contributors to error. The mismatch for actual missions may provide some insight into the errors not captured in this analysis.

This comment is identical to the first one made by referee #3, thus we reproduce below a common answer to both comments.

The point raised here by the referee is very relevant. In the revised manuscript, we have included OCO-2 results in Figures 3, 4, 5 and 6, and in Supplementary Figures S4, S5, S7-10, S12-17,.

We also introduce how we model OCO-2 observations in a Subsection 2.1.

| Line 148 - 154 | The Orbiting Carbon Observatory-2 (OCO-2) has been providing $X_{CO_2}$ observations from SWIR measurements for close to a decade (Taylor et al., 2023). We include this instrument in order to assess how the synthetic results obtained here relate to results obtained from real data. We model OCO-2 observations relying on instrument functions and noise models provided in OCO-2 L1b Science and Standard L2 products of Atmospheric Carbon Observation from Space algorithm version 8 (ACOS, O'Dell et al., 2018). These files are not from the latest v10 version of OCO-2 data, but the v8 to v10 major reprocessing did not include significant changes on instrument parameters (Taylor et al., 2023), so we assess that our input data are acceptable for this synthetic study. |
|---|---|

We finally discuss the obtained XCO2 precision for OCO-2 against the one reported in OCO-2 Standard L2 product in Subsection 5.1:

| Line 643 - 653 | First, OCO-2 shows a noise-only related precision of 0.32 ppm corresponding to DOFs for CO₂-related parameters of 1.97. The OCO-2 results that we obtain are overall consistent with ACOS results for soundings with close band-wise albedo values (see Supplementary Figure S6). Besides, land nadir OCO-2 $X_{CO_2}$ retrievals show an overall 0.77 ppm standard deviation compared to the Total Carbon Column Observing Network (TCCON) validation reference (Taylor et al., 2023). This difference with respect to the theoretical uncertainty computed from Optimal Estimation stems from all the forward and inverse modelling errors that are not accounted for in the retrieval scheme. Thus, |
|---|---|

**this illustrates that the results provided in this study are a lower bound to the actual precisions that these upcoming concepts will have.**

Supplementary Figure S6 (reproduced here in Figure R2.1) provides the XCO2 uncertainty due to noise (field 'xco2_uncert_noise' in L2StdND oco2 files) and CO2-related DOFs. For each albedo model considered in this study, we explored the year 2016 ACOS v8 L2 data downloaded for the work performed in Dogniaux et al. (2021) and averaged precision and DOFs results for soundings that match our albedo models within ±0.05. The error-bars range from the 10th to the 90th percentile of each distribution. Our OCO-2 results have been linearly interpolated to match the average OCO-2 Solar Zenith Angle in the considered ACOS data, and the error bar range from the minimum to the maximum values obtained in our synthetic survey.

We can notice that we obtain DOFs that are quite close to ACOS (a little higher because we fit less geophysical parameters in our state vector), and produce noise-related precision results that are close or lower compared to ACOS. Besides, case-to-case differences are also overall consistent between our results and ACOS (SOL and VEG cases show lower DOFs and higher uncertainties than DES cases). However, as many aspects differ in aerosol models, state vector composition, radiance and noise levels, etc, between the OCO-2 soundings that we average here and our 12 explored observational situations, we refrain from comparing further our results and ACOS', and assess that we find an overall agreement that seems acceptable given the differences between the synthetic evaluation performed here, and the ACOS inverse scheme.

[Figure]

*Figure R2.1. XCO2 uncertainty due to noise and CO2-related DOFs from ACOS v8 L2 data (full colors) and from our synthetic study (light colors). For each albedo model considered in this study, we explored the year 2016 ACOS v8 L2 data downloaded for the work performed in Dogniaux et al. (2021). We averaged the precision field 'xco2_uncert_noise' (in L2StdND oco2 files) and DOFs results for soundings that match our albedo models within ±0.05. The error-bars range from the 10th to the 90th percentile of each distribution. Our OCO-2 results have been linearly interpolated to match the average OCO-2 Solar Zenith Angle in the*

*considered ACOS data for each albedo model, and the error bar range from the minimum to the maximum values obtained in our synthetic survey.*

Specific comments:

1) Lines 210 – use soot and minerals as their aerosols – what justifies these choices? For many parts of the world, these are not representative.

These two models were chosen as possible aerosol types of fine and coarse mode that may pollute the European continent: soot representing here urban pollution and minerals the minerals transported from the Sahara Desert from spring to fall (Papayannis et al, 2008). We do agree with the referee that many different types of aerosols could have been explored, but including a wider range of atmospheric conditions is out of the scope of this study.

We propose to underline this choice better presenting the atmospheric and observational situations explored in this study.

| Lines 242-243 | **As this study focuses on the impact of instrument design parameters on $X_{CO_2}$ retrieval performance, we purposefully limit the number of atmospheric conditions that we include.** We consider 12 atmospheric… |
| --- | --- |

| Lines 264 -266 | **To mimic possible pollution over the European continent we include** fine-mode aerosols, representative of soot, between 0 and 2 km of altitude, and coarse mode aerosols, representative of minerals, between 2 and 4 km of altitude (**this choice is supported by transported desert dust layers over Europe described by (Papayannis et al., 2008).** |
| --- | --- |

How do the absorption and scattering characteristics impact the results?

Here, the optical properties are supposed to be perfectly known, consequently this parameter has not been studied. We do agree with the referee that exploring different aerosol types spanning different optical properties would be valuable. However, exploring a wider range of atmospheric conditions is out of the scope of this study, as doing so would tend towards an actual comprehensive OSSE.

We propose to add a discussion element when presenting the aerosol model repeating that their optical properties are fixed, and some additional conclusion elements mentioning as further work the exploration of a wider variety of atmospheric conditions.

| Lines 842 - 849 | **Given its scope focused on exploring the impact of concept design parameters on $X_{CO_2}$ retrieval performance, this study could not include all the dimensions of a comprehensive mission performance assessment. For example, the accuracy of $X_{CO_2}$** |
| --- | --- |

| | **retrieval has not been studied, and a greater variability of possible atmospheric conditions (different aerosol types, layers, contents, etc., different thermodynamical profiles and $CO_2$ concentration vertical profiles) could be encompassed, as is usually performed in comprehensive Observing System Simulation Experiment. Besides, this work could not also obviously explore the whole extent of possible design parameters (e.g. band-wise variations of spectral sampling ratios, varying wavelength interval for spectral bands, combination of different instruments, etc.) that impact $X_{CO_2}$ retrieval performance, and its implication for anthropogenic plume imaging. These limitations warrant further studies.** |
|---|---|

 2) The paragraph that starts at line 296 discusses Figure 3. The authors use the word "break". I think the changes in slope of these lines is not all that significant, so break is not a good choice of phrasing.  I would suggest a phrase like "change of slope"

We changed the formulation, thank you very much. We replaced "breaks" by "change" in different places of the revised manuscript.

3) Sections 4.1 and 4.2 could have a short introductory paragraph to introduce the structure of the subsections that follow.

We added such short introductory paragraphs.

| Lines 352-354 | **This subsection explores the combined impact of spectral resolution and signal-to-noise ratio on $X_{CO_2}$ retrieval performance. First, we discuss how $X_{CO_2}$ precision and $CO_2$-related degrees of freedom evolve with spectral resolution and signal-to-noise ratio, and then we examine $X_{CO_2}$ vertical sensitivities.** |
|---|---|

| Lines 455-458 | **This subsection explores the combined impact of spectral resolution and band selection on $X_{CO_2}$ retrieval performance. First, we discuss how $X_{CO_2}$ precision and $CO_2$ and non-$CO_2$ related degrees of freedom evolve with spectral resolution and band selection, and then we examine $X_{CO_2}$ vertical sensitivities. Finally, we explore $X_{CO_2}$ sensitivities to a priori misknowledge of interfering geophysical variables, with an eventual focus on aerosol-related parameters.** |
|---|---|

4) Line 330: The starting sentence of this section (4.2.1) is nearly the same text as is used to start section 4.1.1

- On line 270

270 For the atmospheric situation VEG-50o, Figure 3 shows the $X$CO2 precision (or random error and degrees of freedom (hereafter DOFs) as a function of both the resolving power $\lambda/\Delta\lambda$

and the signal-to-noise ratio (SNR) for CVAR, and for the exact CO2M, MicroCarb and NanoCarb concepts (results for exactly-defined concepts are discussed in Sect. 5)

- Line 330

For the atmospheric situation VEG-50o[…], Figure 5 shows the $XCO2$ precision and DOFs as a function of both the resolving power $\lambda/\Delta\lambda$ and spectral band selection for CVAR (with SNR fixed at its reference value), and for the exact CO2M, MicroCarb and NanoCarb concepts (results for exactly-defined concepts are discussed in Sect. 5).

- To address this, Section 4.2.1 could have a sentence to first introduce the focus of the analysis. Perhaps "In this section we assess the impacts of changing the spectral bands". (and section 4.1.1 could be introduced with " Here we look at SNR impacts on precisions and DOF.

We added two small sentences as advised at the beginning of each section.

| Line 359 | **Here, we assess the impact of varying spectral resolution and signal-to-noise ratio.** For the atmospheric situation VEG-50º, Figure 3 shows the $X_{CO_2}$ precision (or random error) and degrees… |
| --- | --- |

| Line 460 | **Here, we assess the impact of varying the spectral resolution and band selection.** For the atmospheric situation VEG-50º (results for other situations are given in the Supplements), Figure 5 shows the $X_{CO_2}$ precision and DOFs as a function of both |
| --- | --- |

5) Lines 382 and following: I find this language to be very convoluted, and suggest a rewrite.

Currently "While methodologies are hardly comparable (because this study is only based on synthetic simulations), both works agree that a sharp change in how $XCO2$ precision evolves with resolving power is to be expected around $\lambda/\Delta\lambda = 1000 – 2000$, when solely using the 1.6 or 2.05 μm CO2 bands"

Suggest:

"While methodologies are hardly comparable (because this study is only based on synthetic simulations), both works agree that the $XCO2$ precision and resolving power relationship has a change of characteristic around $\lambda/\Delta\lambda = 1000 – 2000$, when solely using the 1.6 or 2.05 μm CO2 bands"

We followed the referee's suggestions to reformulate this sentence (New text lines 521-524).

6) Figure 11 – I can not differentiate the colors of MicroCarb B1234 and NanoCarb comp.

We change the linestyle of NanoCarb comb to 'dotted' to help better differentiate them.

7) Line 608 – The phrase "more easily gained" implies that we just need to make higher SNR instruments and we can easily get better precision. But this paper just studies the sensitivities. I would suggest rephrasing to "Overall, precision is more sensitive to SNR improvements than spectral resolution improvements."

We followed the referee's advice in the revised manuscript. In addition, comments from Referee #1 made us revise this very general statement that is not true for all magnitudes of SNR and resolving power changes.

| Lines 392-394 | Hence, it appears that $X_{CO_2}$ precision **is more sensitive to SNR** improvements rather than through resolving power improvements**, for large improvements of two orders of magnitude centred on $CO_2M$ instrument characteristics.** |
|---|---|

| Lines 793-794 | Overall, **for these large changes of about two orders of magnitude**, precision is **more sensitive to** SNR improvement than to spectral resolution improvements. |
|---|---|

Editorial comments:

Line 525 – the word Temperature is capitalized mid-sentence.

We fixed this mistake, thank you very much.

Lines 673, 674, 688, 716, 719 and 720 (and maybe others) – formatting issues in the references – looks like latex formats not properly converted???

Indeed, the Mendely Microsoft Word plugin gave us a hard time building the reference from bibtex files, and so does the Zotero plugin as well.

We fixed all the LaTeX-related typos by hand in the revised manuscript.

**Citation**: https://doi.org/10.5194/amt-2023-233-RC2

**References**

Papayannis, A., et al. (2008), Systematic lidar observations of Saharan dust over Europe in the frame of EARLINET (2000–2002), J. Geophys. Res., 113, D10204, doi:10.1029/2007JD009028.

Dogniaux, M., Crevoisier, C., Armante, R., Capelle, V., Delahaye, T., Cassé, V., De Mazière, M., Deutscher, N. M., Feist, D. G., Garcia, O. E., Griffith, D. W. T., Hase, F., Iraci, L. T., Kivi, R., Morino, I., Notholt, J., Pollard, D. F., Roehl, C. M., Shiomi, K., Strong, K., Té, Y., Velazco, V. A., and Warneke, T.: The Adaptable 4A Inversion (5AI): description and first $X_{CO_2}$ retrievals from Orbiting Carbon Observatory-2 (OCO-2) observations, Atmos. Meas. Tech., 14, 4689–4706, https://doi.org/10.5194/amt-14-4689-2021, 2021.

---

## Author Comment (AC3)

We are grateful to the referee for this feedback and comments. Our answers are given in red between the referee's text.

**General comments**

In this paper, the authors perform a quantitative study assessing the estimated performance of a hypothetical shortwave infrared (SWIR) $CO_2$ satellite instrument, considering the impact of a range of instrument design parameters: spectral resolution, signal-to-noise ratio (SNR), and spectral band selection. They achieve this by applying an optimal estimation retrieval algorithm to synthetic spectra generated assuming a wide range of fictitious instrument concepts, defined by varying each of these parameters, and a number of different observation scenarios. In addition, they apply the same performance assessment framework to some ready-defined future mission concepts – MicroCarb, CO2M, and NanoCarb – providing useful context for the hypothetical concept assessment study. This paper is timely given the wide interest in new methodologies for measuring $CO_2$ emissions, driven by the need to independently verify Paris Agreement objectives, which are likely to include satellite remote sensing as a significant component. There are some particularly interesting conclusions which should help inform the conception and design of future SWIR $CO_2$ satellite missions, namely the relative importance of improving SNR vs. resolving power in order to improve $XCO_2$ precision, the importance of including an $O_2$ absorption band in a mission concept to account for aerosol absorption, and the sensitivity of low SNR and resolving power instrument concepts to a priori mis-knowledge. I think that this paper is suitable for publication in Atmospheric Measurement Techniques, and have a few suggestions for improvements which will hopefully help strengthen the paper's conclusions further.

**Specific comments**

1. As mentioned above, the inclusion of ready-defined mission concepts provides useful context for the fictitiously varying CO2M (CVAR) concept study. I think that the paper overall would benefit by also considering an existing mission – OCO-2 for example – along with the ready-defined future missions already included. This would provide additional context for the CVAR study by comparing their performances alongside the current "state-of-the-art", whilst also demonstrating that the assumed observation scenarios and the forward and inverse setups produce realistic results when compared with real observational data;

This comment is identical to the second one made by referee #2, thus we reproduce below a common answer to both comments.

The point raised here by the referee is very relevant. In the revised manuscript, we have included OCO-2 results in Figures 3, 4, 5 and 6, and in Figures S4, S5, S7-10, S12-17, etc.

We also introduce how we model OCO-2 observations in a Subsection 2.1.

| Line 148 - 154 | **The Orbiting Carbon Observatory-2 (OCO-2) has been providing $X_{CO_2}$ observations from SWIR measurements for close to a decade (Taylor et al., 2023). We include this instrument in order to assess how the synthetic results obtained here relate to results obtained from real data. We model OCO-2 observations relying on instrument functions and noise models provided in OCO-2 L1b** |

| | **Science and Standard L2 products of Atmospheric Carbon Observation from Space algorithm version 8 (ACOS, O'Dell et al., 2018). These files are not from the latest v10 version of OCO-2 data, but the v8 to v10 major reprocessing did not include significant changes on instrument parameters (Taylor et al., 2023), so we assess that our input data are acceptable for this synthetic study.** |
|---|---|

We finally discuss the obtained XCO2 precision for OCO-2 against the one reported in OCO-2 Standard L2 product in Subsection 5.1:

| Line 643 - 653 | **First, OCO-2 shows a noise-only related precision of 0.32 ppm corresponding to DOFs for $CO_2$-related parameters of 1.97. The OCO-2 results that we obtain are overall consistent with ACOS results for soundings with close band-wise albedo values (see Supplementary Figure S6). Besides, land nadir OCO-2 $X_{CO_2}$ retrievals show an overall 0.77 ppm standard deviation compared to the Total Carbon Column Observing Network (TCCON) validation reference (Taylor et al., 2023). This difference with respect to the theoretical uncertainty computed from Optimal Estimation stems from all the forward and inverse modelling errors that are not accounted for in the retrieval scheme. Thus, this illustrates that the results provided in this study are a lower bound to the actual precisions that these upcoming concepts will have.** |
|---|---|

Supplementary Figure S6 (reproduced here in Figure R2.1) provides the XCO2 uncertainty due to noise (field 'xco2_uncert_noise' in L2StdND oco2 files) and CO2-related DOFs. For each albedo model considered in this study, we explored the year 2016 ACOS v8 L2 data downloaded for the work performed in Dogniaux et al. (2021) and averaged precision and DOFs results for soundings that match our albedo models within ±0.05. The error-bars range from the 10th to the 90th percentile of each distribution. Our OCO-2 results have been linearly interpolated to match the average OCO-2 Solar Zenith Angle in the considered ACOS data, and the error bar range from the minimum to the maximum values obtained in our synthetic survey.

We can notice that we obtain DOFs that are quite close to ACOS (a little higher because we fit less geophysical parameters in our state vector), and produce noise-related precision results that are close or lower compared to ACOS. Besides, case-to-case differences are also overall consistent between our results and ACOS (SOL and VEG cases show lower DOFs and higher uncertainties than DES cases). However, as many aspects differ in aerosol models, state vector composition, radiance and noise levels, etc, between the OCO-2 soundings that we average here and our 12 explored observational situations, we refrain from comparing further our results and ACOS', and assess that we find an overall agreement that seems acceptable given the differences between the synthetic evaluation performed here, and the ACOS inverse scheme.

[Figure]

*Figure R2.1. XCO2 uncertainty due to noise and CO2-related DOFs from ACOS v8 L2 data (full colors) and from our synthetic study (light colors). For each albedo model considered in this study, we explored the year 2016 ACOS v8 L2 data downloaded for the work performed in Dogniaux et al. (2021). We averaged the precision field 'xco2_uncert_noise' (in L2StdND oco2 files) and DOFs results for soundings that match our albedo models within ±0.05. The error-bars range from the $10^{th}$ to the $90^{th}$ percentile of each distribution. Our OCO-2 results have been linearly interpolated to match the average OCO-2 Solar Zenith Angle in the considered ACOS data for each albedo model, and the error bar range from the minimum to the maximum values obtained in our synthetic survey.*

I think some further justification/clarification would be useful for the atmospheric situations used in the study. For example, are the temperature and water vapour profiles from the TGIR climatology representative of the current climate? Similarly, I think it would strengthen the conclusions if a realistic profile of $CO_2$ concentration were used instead of a constant profile, especially given that the study considers the vertical sensitivity of the instrument concepts;

The thermodynamic (temperature, water vapor) atmospheric profile used in this work and taken from TIGR is identical to the one used for the initial NanoCarb L2 performance assessment in Dogniaux et al. (2022). The TIGR profiles are appropriate to describe the current climate in the context of spaceborne greenhouse gas monitoring. For example, they are used to build the training dataset of the neural networks used to retrieve mid-tropospheric columns CO2 and CH4 from IASI thermal infrared observations (Crevoisier et al, 2009a,b). These are included in the CAMS greenhouse gas analysis running from 2003 to 2020 (Agustí-Panareda et al., 2023).

Regarding realistic CO2 concentration profiles, we agree with the referee that the study could have included various vertical CO2 profiles, especially with and without anthropogenic enhancement of CO2 concentration in the lower layers. However, as this study is dedicated on evaluating the impact of instrument design parameters on XCO2 retrieval performance, and as it includes NanoCarb which is still in early stages of L2 performance evaluations, we decided to stick to the simple vertically constant CO2 profile for with which we performed the initial initial NanoCarb L2 performance assessment in Dogniaux et al. (2022). Clearly

stating further steps of including more realistic CO2 profiles should be included in the conclusions, and it was not in the original manuscript. We did so in the revised version.

| Lines 842 - 849 | **Given its scope focused on exploring the impact of concept design parameters on $X_{CO_2}$ retrieval performance, this study could not include all the dimensions of a comprehensive mission performance assessment. For example, the accuracy of $X_{CO_2}$ retrieval has not been studied, and a greater variability of possible atmospheric conditions (different aerosol types, layers, contents, etc., different thermodynamical profiles and $CO_2$ concentration vertical profiles) could be encompassed, as is usually performed in comprehensive Observing System Simulation Experiment. Besides, this work could not also obviously explore the whole extent of possible design parameters (e.g. band-wise variations of spectral sampling ratios, varying wavelength interval for spectral bands, combination of different instruments, etc.) that impact $X_{CO_2}$ retrieval performance, and its implication for anthropogenic plume imaging. These limitations warrant further studies.** |
| --- | --- |

2. Whilst this study does not explicitly consider spatial resolution, I think it would be worth commenting on the implications of some of the conclusions on the feasibility of CO2 imaging concepts, which trade off reduced SNR and/or resolving power in favour of high spatial resolution in order to be able to quantify emissions from ever-smaller plumes of CO2 emitted by point sources. To pick one example from the results in Section 4, Figure 9 shows how concepts with low resolving power would be quite sensitive to a priori mis-knowledge of aerosol optical depths, depending on the spectral band selected and whether an O2 absorption band is incorporated into the instrument concept.

We agree with the referee that such points are indeed interesting, and were discussed in Sect. 4.2.3 of the original manuscript. They could have been included in the conclusions as well. We adjusted the conclusions in the revised manuscript to reflect this discussion point as well.

| Lines 805-809 | **These results highlight how the precise (and accurate to some extent) retrieval of $X_{CO_2}$ from SWIR observations relies on the amount of information carried by these observations. Reducing spectral resolution and/or the number of spectral bands to improve spatial resolution increases errors that may be removed when imaging local relative enhancements of $X_{CO_2}$. However, they may still hamper absolute $X_{CO_2}$ retrievals in plume-free scenes, thus potentially making these observations hardly useful for anything else than anthropogenic emission imaging.** |
| --- | --- |

Further investigation looking at the ability of SWIR hyperspectral imagers to image emissions plumes and infer CO2 emission rates, using the performance assessment framework described here across a range of instrument parameters including spatial resolution would be very interesting, but I appreciate that would be beyond the scope of this study.

We agree with the referee that including a plume-imaging angle to this study would have been a further interesting angle, but it would indeed have extended this work far beyond its intended scope. However, these further steps are relevant to include as perspective in the conclusions, that were not complete in the original manuscript. We extended them in the revised manuscript.

| Lines 842 - 849 | **Given its scope focused on exploring the impact of concept design parameters on $X_{CO_2}$ retrieval performance, this study could not include all the dimensions of a comprehensive mission performance assessment. For example, the accuracy of $X_{CO_2}$ retrieval has not been studied, and a greater variability of possible atmospheric conditions (different aerosol types, layers, contents, etc., different thermodynamical profiles and $CO_2$ concentration vertical profiles) could be encompassed, as is usually performed in comprehensive Observing System Simulation Experiment. Besides, this work could not also obviously explore the whole extent of possible design parameters (e.g. band-wise variations of spectral sampling ratios, varying wavelength interval for spectral bands, combination of different instruments, etc.) that impact $X_{CO_2}$ retrieval performance, and its implication for anthropogenic plume imaging. These limitations warrant further studies.** |
| --- | --- |

**Technical corrections**

Line 232: replace "Its" with "It is";

There is no spelling mistake here. We are referring to the a posteriori covariance matrix *of* the state vector.

Line 243: replace "degree" with "degrees";

We fixed this mistake, thank you.

Line 372: please provide a reference for the "usual" hypothesis that aerosol properties are fixed across spectral bands;

For example, ACOS uses fixed aerosol optical properties, we added a reference.

| Line 513 | This result is made possible by the usual **(see OCO-2 processing algorithm ACOS for example, O'Dell et al., 2018)** hypothesis of fixed aerosol optical properties, which enables sharing optical path information across spectral bands. |
| --- | --- |

Line 493: replace "MC123" with "MC234".

We fixed the mistake, thank you very much.

**References**

Crevoisier, C., Nobileau, D., Fiore, A. M., Armante, R., Chédin, A., and Scott, N. A.: Tropospheric methane in the tropics – first year from IASI hyperspectral infrared observations, Atmos. Chem. Phys., 9, 6337–6350, https://doi.org/10.5194/acp-9-6337-2009, 2009.

Crevoisier, C., Chédin, A., Matsueda, H., Machida, T., Armante, R., and Scott, N. A.: First year of upper tropospheric integrated content of CO2 from IASI hyperspectral infrared observations, Atmos. Chem. Phys., 9, 4797–4810, https://doi.org/10.5194/acp-9-4797-2009, 2009.

Agustí-Panareda, A., Barré, J., Massart, S., Inness, A., Aben, I., Ades, M., Baier, B. C., Balsamo, G., Borsdorff, T., Bousserez, N., Boussetta, S., Buchwitz, M., Cantarello, L., Crevoisier, C., Engelen, R., Eskes, H., Flemming, J., Garrigues, S., Hasekamp, O., Huijnen, V., Jones, L., Kipling, Z., Langerock, B., McNorton, J., Meilhac, N., Noël, S., Parrington, M., Peuch, V.-H., Ramonet, M., Razinger, M., Reuter, M., Ribas, R., Suttie, M., Sweeney, C., Tarniewicz, J., and Wu, L.: Technical note: The CAMS greenhouse gas reanalysis from 2003 to 2020, Atmos. Chem. Phys., 23, 3829–3859, https://doi.org/10.5194/acp-23-3829-2023, 2023.